# pH and Redox-Dual Sensitive Chitosan Nanoparticles Having Methyl Ester and Disulfide Linkages for Drug Targeting against Cholangiocarcinoma Cells

**DOI:** 10.3390/ma15113795

**Published:** 2022-05-26

**Authors:** Ju-Il Yang, Hye Lim Lee, Je-Jung Yun, Jungsoo Kim, Kyoung-Ha So, Young-IL Jeong, Dae-Hwan Kang

**Affiliations:** 1Department of Medical Science, School of Medicine, Pusan National University, Busan 50612, Korea; yangjuil@outlook.kr; 2Department of Internal Medicine, Yangsan Hospital, Pusan National University, Busan 50612, Korea; 3Research Institute of Convergence of Biomedical Science and Technology, Yangsan Hospital, Pusan National University, Busan 50612, Korea; roasua@hanmail.net (H.L.L.); jskimpnuh@naver.com (J.K.); 4Research Center for Environmentally Friendly Agricultural Life Science, Jeonnam Bioindustry Foundation, Gokseong-gun 57509, Korea; jjyoung4@hanmail.net; 5School of Chemical and Biological Engineering, Institute of Chemical Processes, Seoul National University, Seoul 08826, Korea

**Keywords:** redox sensitive, acidic pH sensitive, nanoparticles, cholangiocarcinoma, drug targeting

## Abstract

The aim of this study is to prepare pH- and redox-sensitive nanoparticles for doxorubicin (DOX) delivery against DOX-resistant HuCC-T1 human cholangiocarcinoma (CCA) cells. For this purpose, L-histidine methyl ester (HIS) was attached to chitosan oligosaccharide (COS) via dithiodipropionic acid (abbreviated as ChitoHISss). DOX-incorporated nanoparticles of ChitoHISss conjugates were fabricated by a dialysis procedure. DOX-resistant HuCC-T1 cells were prepared by repetitive exposure of HuCC-T1 cells to DOX. ChitoHISss nanoparticles showed spherical morphology with a small diameter of less than 200 nm. The acid pH and glutathione (GSH) addition induced changes in the size distribution pattern of ChitoHISss nanoparticles from a narrow/monomodal distribution pattern to a wide/multimodal pattern and increased the fluorescence intensity of the nanoparticle solution. These results indicate that a physicochemical transition of nanoparticles can occur in an acidic pH or redox state. The more acidic the pH or the higher the GSH concentration the higher the drug release rate was, indicating that an acidic environment or higher redox states accelerated drug release from ChitoHISss nanoparticles. Whereas free DOX showed decreased anticancer activity at DOX-resistant HuCC-T1 cells, DOX-incorporated ChitoHISss nanoparticles showed dose-dependent anticancer activity. Intracellular delivery of DOX-incorporated ChitoHISss nanoparticles was relatively increased at an acidic pH and in the presence of GSH, indicating that DOX-incorporated ChitoHISss nanoparticles have superior acidic pH- and redox-sensitive behavior. In an in vivo tumor xenograft model, DOX-incorporated ChitoHISss nanoparticles were specifically delivered to tumor tissues and then efficiently inhibited tumor growth. We suggest that ChitoHISss nanoparticles are a promising candidate for treatment of CCA.

## 1. Introduction

Cholangiocarcinoma (CCA), which is a malignant tumor in the epithelium of the biliary tract, is frequently shown to have poor prognosis, and the incidence rate of CCA is increasing worldwide [1,2,3]. Since early diagnosis of CCA is difficult and then is frequently diagnosed in an advanced stage, surgical resection, which is a curative option, is practically impossible [4,5]. Except for surgical resection, treatment options such as stent displacement, radiotherapy, chemotherapy, and immunotherapy have been used to try to treat CCA in the last several decades [6,7,8,9]. Among them, chemotherapy has frequently been considered to improve the survivability and life quality of CCA patients [10,11,12,13]. Clinical trials of chemotherapeutic agents including cisplatin, epirubicin, 5-fluorouracil, and gemcitabine have tried to manage biliary tract adenocarcinoma with manageable toxicity against patients [10]. Kim et al. also reported that a combination of gemcitabine and cisplatin was tolerable for patients with inoperable biliary tract cancer and showed modest response rates [11]. A combination of cisplatin and gemcitabine is believed to be a synergistic candidate for biliary tract cancer compared to single treatment [12]. Wang et al. reported that a hepatic arterial infusion of oxaliplatin and 5-fluorouracil is beneficial to controlling tumor progression, the survivability of patients, and toxicity for advanced perihilar cholangiocarcinoma (PCC) [13]. It was also reported that chemotherapy followed by radiation therapy has a beneficial effect against unresectable perihilar CCA [14]. However, most of the treatment regimens, such as chemotherapy and radiotherapy, have no benefit to the survivability of patients [15,16]. From these points of view, targeted therapy using molecular-targeted agents has tried to improve the therapeutic efficacy and survival period of CCA patients [16,17]. Even though molecular-targeted agents have been suggested as a promising candidate for targeted therapy, their efficacy still provides insignificant benefit in the survivability of CCA patients [16,17,18,19]. The multi-drug resistance (MDR) of CCA against conventional chemotherapeutic agents and/or molecular-targeted agents is also problematic for improvement of therapeutic responses and patient survivability [19,20,21]. For example, Chakrabarti et al. reported that the drug-resistant problem of fibroblast growth factor receptor (FGFR) inhibitors is problematic and has to be solved for future trials [19]. Massa et al. also reported that paclitaxel-incorporated albumin nanoparticles have a benefit in overcoming MDR and then delaying tumor growth/vasculature [21]. Therefore, novel anticancer agents based on nanoparticles should be developed to overcome MDR of CCA.

Nanoscale-based carriers such as liposomes, nanoparticles, and polymeric micelles have been extensively investigated for the tumor-specific delivery of bioactive agents [22,23,24,25,26]. Nanoparticles are frequently employed to deliver anticancer drugs against solid tumor because they have a large surface area for easy modification, a small diameter to avoid the reticuloendothelial system, and structural peculiarity for payload hydrophobic drugs [27]. In particular, the biochemical and physiological status of the tumor microenvironment is quite different compared to normal tissues [28]. Physiological peculiarities of tumor tissues are an acidic pH environment, vascularization, elevated levels of reduction/oxidation (redox) potential, expression of various molecular receptors, changes in perfusion rate, leaky blood vessels, etc. [28,29,30]. The acid pH of the tumor microenvironment has been applied to control the drug delivery behavior of nanocarriers in tumor tissues [31]. Du et al. reported that properties of nanoparticles can be changed to adapt to the acidic pH of the tumor extracellular environment and intracellular environment [32]. They argued that nanoparticles with an acid-cleavable group have sensitivity against the acidic pH of the tumor microenvironment and then improve drug-delivery capacity. It was also reported that glutathione (GSH) levels in the tumor microenvironment are significantly higher than normal tissues [33]. The elevated levels of GSH in tumor tissues are frequently associated with drug-resistance problems [34]. Sun et al. reported that the DOX release rate from polymeric micelles with disulfide linkages is accelerated in the intracellular compartment of tumor cells because the intracellular GSH level in tumor cells is significantly higher than the extracellular GSH level and the disulfide bond is able to be disintegrated by GSH [35]. Park et al. also reported that polymer nanoparticles with disulfide linkages were cleaved by GSH and cancer cell viability was efficiently inhibited through redox-sensitive delivery of anticancer drugs against cancer cells [36].

In this study, we synthesized chitosan-histidine conjugates using disulfide linkage (ChitoHISss) and fabricated nanoparticles to overcome MDR of CCA cells. In addition, doxorubicin (DOX)-incorporated ChitoHISss nanoparticles were fabricated for pH- and redox-sensitive delivery of DOX against HuCC-T1 human cholangiocarcinoma cells. L-histidine methyl ester and cystamine were employed to endow pH and redox-sensitivity to chitosan nanoparticles since histidine has an acidic sensitivity and cystamine can be cleaved by GSH. DOX-resistant CCA cells were prepared for the investigation of the drug-delivery potential of ChitoHISss nanoparticles.

## 2. Materials and Methods

### 2.1. Chemicals

Chitosan oligosaccharide (COS) was purchased from Tokyo Chemical Industry (TCI) Co., Ltd. (Tokyo, Japan). Doxorubicin (DOX) was purchased from LC Labs^®^ Co. (Woburn, MA, USA). Chlorin e6 (Ce6) was obtained from Frontier Sci. Co. (Logan, UT, USA). Pyrene, L-histidine methyl ester dihydrochloride (HIS), 3,3′-dithiodipropionic acid di (N-hydroxysuccinimide ester) (DTP-NHS), L-glutathione reduced (GSH), tribromoethanol (avertin), triethylamine (TEA), 3-(4,5-dimethyl-2-thiazolyl)-2, 5-diphenyl-2H-tetrazolium bromide (MTT), dimethyl sulfoxide (DMSO), and methanol (MeOH) were purchased from Sigma Aldrich Chem. Co. (St. Louis, MO, USA). Dialysis membranes (molecular weight cutoffs (MWCO): 1000 and 2000 Da) were purchased from Spectrum Labs., Inc. (Rancho Dominguez, CA, USA). Organic solvents such as DMSO and MeOH were used in an ultrapure grade.

### 2.2. Synthesis of ChitoHISss Conjugates

HIS (242.1 mg, 1 mM) with an equal amount of TEA dissolved in 10 mL DMSO was mixed with 404.4 mg of DTP-NHS. This reaction was stirred for 6 h. COS (400 mg) was dissolved in a 10 mL DMSO/water mixture (DMSO: water = 4:1) and then this solution was mixed with HIS/DTP-NHS solution. Following this, the mixtures were magnetically stirred for 24 h, and then the resulting solution was transferred to a dialysis membrane (MWCO: 2000 Da). This was dialyzed against 3 L distilled water to remove organic solvent, unreacted chemicals, and byproducts. Water was exchanged every 3 h for 48 h to avoid saturation of solvents, and then this solution was freeze-dried for 3 days to obtain a solid. The resulting products were named ChitoHISss conjugates. The yield of ChitoHISss conjugates was evaluated by mass measurement, and the yield was approximately 93.2%. Equation for yield = (weight of ChitoHISss conjugates)/(feeding weight of HIS + feeding weight of DTP-NHS).

### 2.3. H Nuclear Magnetic Resonance (NMR) Spectra

A Varian Unity Inova 500 MHz NB high-resolution Fourier transform (FT)-NMR spectrometer (Varian Inc., Santa Clara, CA, USA) was employed to monitor the chemical structure of the conjugates. For analysis, chemicals were dissolved in DMSO or D_2_O/DMSO mixtures and then measured using ^1^H NMR spectra.

### 2.4. Preparation of DOX-Incorporated ChitoHISss Nanoparticles

DOX (5~10 mg) was dissolved in 2 mL DMSO with a similar amount of TEA. ChitoHISss conjugates (40 mg) were dissolved in 5 mL DMSO/water mixture (4/1, *v*/*v*) and then mixed with DOX solution. This solution was magnetically stirred for 10 min and then dropped into 10 mL distilled water. The resulting solution was introduced into a dialysis membrane (MWCO: 2000 g/mol) and then dialyzed against 1 L water for 1 day. Water was exchanged in 2–3 h intervals for 24 h and then dialyzed solution was lyophilized or used for analysis.

To evaluate drug contents, the volume of dialyzed solution was adjusted to 40 mL using distilled water. After that, 5 mL of this solution was diluted with DMSO more than 10 times. The DOX concentration was measured at 479 nm with a UV spectrophotometer (UV-1601 UV-VIS spectrophotometer, Shimadzu, Kyoto, Japan). The drug content was calculated as follows: drug content (*w*/*w*) = (DOX weight in the nanoparticles/nanoparticle weight) × 100; loading efficiency (*w*/*w*) = (DOX weight in the nanoparticles/feeding weight of DOX) × 100.

Empty nanoparticles were prepared with the same procedure described above in the absence of DOX.

### 2.5. Preparation of Ce6-Incorporated ChitoHISss Nanoparticles

Fluorescent dye Ce6 was used for the study of fluorescence characteristics and animal imaging of ChitoHISss nanoparticles. Ce6 (2 mg) was dissolved in 1 mL DMSO. ChitoHISss conjugates (20 mg) were dissolved in 4 mL DMSO/water mixture (4/1, *v*/*v*) and then mixed with Ce6 solution. These mixtures were magnetically stirred for 10 min, dropped into 5 mL distilled water, and introduced into a dialysis membrane (MWCO: 2000 g/mol) for dialysis. The dialysis procedure was performed against 1 L water for 1 day with an exchange of water at 2–3 h intervals. The resulting solution was adjusted to 20 mL. This solution was used to measure Ce6 content in the nanoparticles using a fluorescence spectrophotometer as follows: 1 mL of Ce6-incorporated ChitoHISss nanoparticle solution was diluted with DMSO more than 10 times to measure the Ce6 concentration in the nanoparticles using a fluorescence spectrophotometer (excitation wavelength: 407, emission wavelength: 664 nm) (RF-5301PC spectrofluorophometer, Kyoto, Japan). Free Ce6 was dissolved in DMSO for comparison.
Ce6 contents (wt.%) = (Ce6 weight/total weight of nanoparticles)/100.
Ce6 contents in the nanoparticles were approximately 8.9% (*w*/*w*).

### 2.6. Transmission Electron Microscope (TEM)

TEM (H-7600, Hitachi Instruments Ltd., Tokyo, Japan) was employed to observe the morphology of ChitoHISss nanoparticles. Aqueous nanoparticle solution was dropped onto the carbon film-coated grid and then dried at room temperature. TEM observation was carried out at 80 kV.

### 2.7. Analysis of Particle Size Distribution

Zetasizer Nano-ZS^®^ (Malvern, Worcestershire, UK) was employed to measure particle size distribution. The nanoparticle concentration in the distilled water was adjusted to 0.1% (*w*/*w*) and measured at 20 °C.

### 2.8. Fluorescence Spectrophotometer Measurement of Nanoparticle Solution

The nanoparticle solution was measured with a fluorescence spectrofluorophotometer (Shimadzu RF-5301PC spectrofluorophometer, Kyoto, Japan) to analyze pH and redox sensitivity. Ce6-incorporated nanoparticles were reconstituted in the phosphate-buffered saline (PBS, 0.01 M, pH 7.4; Ce6 concentration, 0.1 mg/mL PBS). For pH sensitivity study, the pH of this solution was adjusted with 0.1 N HCl or 0.1 N NaOH solution. For redox sensitivity, GSH was added to this solution and then incubated for 3 h at 37 °C. Following this, fluorescence emission spectra were measured between 500 nm and 800 nm (excitation wavelength: 400 nm). Fluorescence images of same solution were observed with a Maestro 2 small animal imaging instrument (Cambridge Research and Instrumentation Inc., Woburn, MA, USA).

To study the nano-aggregation behavior of ChitoHISss conjugates, critical aggregation concentration was (CAC) was measured with a fluorescence spectrophotometer (Shimadzu RF-5301PC spectrofluorophometer, Kyoto, Japan) using pyrene. A total of 100 μL of pyrene solution in acetone was pipetted into a vial, acetone was evaporated in room temperature, and then 10 mL aqueous nanoparticle solution was poured into the vial (final concentration of pyrene: 6.0 × 10^−7^ M). These solutions were equilibrated at 65 °C for 3 h following with cooling at room temperature for 2 h. Fluorescence excitation spectra of these solutions were measured at 300 nm~350 nm of the emission wavelength (emission wavelength, 390 nm; excitation and emission bandwidths, 1.5 nm and 1.5 nm).

### 2.9. Drug Release from Nanoparticles

The concentration of aqueous nanoparticle solution was adjusted to 1 mg/mL with PBS, and then 5 mL this solution was introduced into the dialysis membrane (MWCO = 2000 g/mol). This was put into a Falcon^®^ tube (Thermo Fisher Sci., Co., Waltham, MA, USA) with 45 mL PBS. To study the redox sensitivity of the nanoparticles, GSH was added to this solution. For pH sensitivity, the pH of the media was changed to an acidic pH with 0.1 N HCl solution. This was then incubated at 37 °C and 100 rpm in a shaker incubator (SI-600R, Jeiotech Co., Daejeon, Korea). Whole media were taken to analyze the DOX concentration. The DOX concentration in the media was measured with a UV-VIS spectrophotometer at 479 nm with a UV spectrophotometer (UV-1601 UV-VIS spectrophotometer, Shimadzu, Kyoto, Japan).

### 2.10. Cell Culture

HuCC-T1 human cholangiocarcinoma cells were obtained from Health Science Research Resources Bank (Osaka, Japan). CCD986Sk human skin fibroblast cells were purchased from the Korean Cell Line bank (Seoul, Korea). HuCC-T1 cells were maintained in RPMI1640 medium (Gibco, Grand Island, NY, USA) and supplemented with 10% heat-inactivated fetal bovine serum (FBS) (Invitrogen, Waltham, MA, USA) and 1% penicillin/streptomycin at 37 °C in a 5% CO_2_ incubator. CCD986Sk cells were cultured in IMDM (Gibco, Grand Island, NY, USA) medium supplemented with 10% FBS (Invitrogen, Waltham, MA, USA) and 1% penicillin/streptomycin.

DOX-resistant HuCC-T1 cells were prepared as follows: DOX in serum-free media was treated to HuCC-T1 cells for 1 h and then the media were discarded. Cells were washed with PBS and then fresh growth media were added. These were further incubated for 2 days. Temporary treatment of DOX to HuCC-T1 cells was repeated three times with the same concentration. To increase DOX resistance, the treatment concentration of DOX was gradually increased from 0.0001 µg/mL to 0.1 µg/mL over 3 months.

### 2.11. Anticancer Activity of DOX-Incorporated ChitoHISss Nanoparticles against DOX-Resistant HuCC-T1 Cells

To assess the anticancer activity of DOX-incorporated ChitoHISss nanoparticles, HuCC-T1 cells (1 × 10^4^ cells/well) seeded in 96-well plates (SPL Life Sci., Pocheon-si, Gyeonggi-do, Korea) were incubated overnight in 5% CO_2_ at 37 °C. For DOX treatment, free DOX was dissolved in DMSO and diluted with media. For nanoparticle treatment, an aqueous solution of DOX-incorporated ChitoHISss nanoparticles were sterilized with a 1.2 µm syringe filter (Minisart^®^ Syringe filter, Sartorius AG, Göttingen, Land Niedersachsen, Germany) and then diluted with media. DMSO (final concentration: 0.5 % (*v*/*v*)) was used for the control treatment. Cells were exposed to free DOX, DOX-incorporated ChitoHISss nanoparticles, and empty nanoparticles for 1 or 2 days. Cell viability was evaluated with an MTT proliferation assay. MTT solution (30 µL, 5 mg/mL in PBS) was added to cells in 96 wells and then incubated for 3 h. Supernatants were discarded, DMSO (100 µL) was added to dissolve viable cells, and then the absorbance was measured at 570 nm using an Infinite M200 pro microplate reader. Each measurement was average ± standard deviation (S.D.) from eight wells of 96-well plates.

### 2.12. Observation of Cells with Fluorescence Microscope

For fluorescence observation of cells, 3 × 10^5^ HuCC-T1 cells were seeded in 6 wells with cover glass. These were treated with free DOX or DOX-incorporated ChitoHISss nanoparticles for 60 min. After that, cells were washed with PBS, fixed with 4% paraformaldehyde for 15 min, washed again with PBS, and then immobilized with mounting solution (Immunomount, Thermo Electron Co., Pittsburgh, PA, USA). The cells were observed with a fluorescence microscope (Eclipse 80i; Nikon, Tokyo, Japan). Each measurement from fluorescence observations and analysis was repeated at least three times and then presented as an average image.

### 2.13. In Vivo Animal Study

A tumor xenograft model of HuCC-T1 cells was prepared to study the antitumor activity of DOX-incorporated ChitoHISss nanoparticles. 1 × 10^7^ HuCC-T1 cells were subcutaneously injected into the backs of male nude mice. Male nude mice (4–5 weeks old, 20–25 g) (Orient, Seongnam, Gyeonggido, Korea) were used for animal study. Five male mice were used for each group. Free DOX solution, DOX-incorporated ChitoHISss nanoparticles, and empty nanoparticles were intravenously (i.v.) injected via the tail vein of mice when the diameter of the tumor mass reached approximately 4–5 mm. The injection volume was 100 µL. The treatment dose with DOX was adjusted to 10 mg/kg. Five mice were used for each group. Tumor volume and body weight were measured in intervals of 5 days. The day of drug injection was determined as the first day. The largest and smallest diameters of the tumor were measured and then the tumor volume was evaluated using the following formula: V = (a × [b]^2^)/2. a, largest diameter; b, smallest diameter. All results are expressed as average ± S.D. from five mice.

For fluorescence imaging of the animals, 1 × 10^7^ HuCC-T1 cells were subcutaneously injected into the backs of male nude mice. When the diameter of the tumor mass became larger than 6 mm, Ce6-incorporated nanoparticles were intravenously (i.v.) injected into the tail vein (10 mg/kg as a Ce6 concentration) of the mice. The injection volume was 100 µL. One day later, the mice were anesthetized with avertin to observe the fluorescence imaging of the HuCC-T1 tumor. After that, the mice were sacrificed for observation of each organ. A Maestro^TM^ 2 small animal-imaging instrument (Cambridge Research and Instruments, Inc. Woburn, MA, USA) was used for observation of the biodistribution of nanoparticles. Each measurement from fluorescence observations and analysis was repeated at least three times and then presented as an average image.

### 2.14. Statistical Analysis

A one-way analysis of variance (ANOVA) followed by the Tukey test was employed to analyze the statistical significance using GraphPad Prism 9 (GraphPad Software LLC., San Diego, CA, USA). *p* < 0.05 as the minimum of significance was evaluated.

## 3. Results

### 3.1. Synthesis of ChitoHISss Conjugates

To synthesize ChitoHISss copolymer, HIS was conjugated with amine groups of COS to endow pH sensitivity and disulfide linkage was introduced between HIS and COS to endow redox sensitivity. Figure 1 shows the synthesis scheme of ChitoHISss conjugates. As shown in Figure 1a,b, specific peaks of HIS and ethyl protons of DTP-NHS were confirmed at 2~10 ppm and 2.8–3.4 ppm, respectively. The amine group of HIS was conjugated with the one-end NHS group of DTP-NHS to produce HIS-DTP conjugates, as shown in Figure 1c. Specific peaks of HIS and DTP-NHS were confirmed between 2 and 10 ppm, indicating that HIS and TDP conjugates successfully conjugated. HIS-DTP conjugates were attached again with the amine group of COS to make ChitoHISss conjugates, as shown in Figure 1e. ^1^H NMR spectra of COS are shown in Figure 1d. As shown in Figure 1d, specific peaks of glucosamine protons were confirmed at 2.5–5.0 ppm. The acetyl group of COS was also confirmed at 1.7 ppm. As shown in Figure 1e, ChitoHISss conjugates showed specific peaks of COS (c1–c7), HIS (s1, s2, s3), and DTP (s4) at 1.5–5.0 ppm, indicating that HIS-DTP conjugates were successfully conjugated with COS. The yield of the final product was estimated by weight measurement, and the yield was approximately 93.2% (*w*/*w*).

### 3.2. Fabrication and Characterization of DOX-Incorporated ChitoHISss Nanoparticles

Empty and DOX-incorporated nanoparticles using ChitoHISss conjugates were prepared by a dialysis procedure. Since HIS is a hydrophobic moiety in the ChitoHISss conjugates and DOX is also a lipophilic drug, ChitoHISss and DOX can be aggregated as nanoparticles. To confirm nanoparticle formation, TEM was employed to observe nano-aggregates, as shown in Figure 2. As shown in Figure 2a, spherical nanoparticles with a small diameter of less than 200 nm were observed. In the analysis of TEM photo, the average diameter of ChitoHISss nanoparticles was 134.5 ± 18.4 nm (Appendix A). When particle size was measured as shown in Figure 2b, their sizes were 120.5 nm with narrow distribution. Drug content and particle size are summarized in Table 1. As shown in Table 1, a higher drug feeding weight induced a higher drug content, indicating that DOX can be loaded into the nanoparticles through a hydrophobic interaction with the lipophilic segment (HIS) of ChitoHISss. A higher drug content induced a larger particle size, as shown in Table 1.

In particular, ChitoHISss has an amphiphilic property and is able to be aggregated by itself in an aqueous solution. The critical aggregation concentration (CAC) was evaluated to define the nano-aggregation properties, as shown in Figure 2c. The partition of pyrene (6.0 × 10^−7^ M) into the core of the nanoparticles was assessed as fluorescence excitation spectra, and then a red shift of pyrene was observed according to the increase in nanoparticle concentration, as shown in Figure 2c. The (0,0) bands in the excitation spectra of pyrene were compared in the intensity ratio I_337.0_/I_334.0_, as shown in Figure 2c. At fluorescence excitation spectra, a cross-over region was observed between the flat region and the sigmoidal region, as shown in Figure 2c. This region was indicated as a CAC value, and the CAC value was approximately 0.0029 g/L.

To assess the dual sensitivity of ChitoHISss nanoparticles against pH and redox status, pH was adjusted to the acidic pH of aqueous nanoparticle solution and GSH was added to the nanoparticle solution. These solutions were incubated and then changes in particle sizes were measured, as shown in Figure 3. As shown in Figure 3a, ChitoHISss nanoparticles showed small particle sizes of less than 150 nm with a narrow distribution at pH 7.4. However, the particle size and size distribution became larger in diameter and broader in size distribution, respectively, at pH 6.8 (Figure 3b) and 6.0 (Figure 3c), indicating that ChitoHISss nanoparticles swelled in the acidic pH due to the HIS moiety in the ChitoHISss conjugates. When GSH was added to the nanoparticle solution and then incubated, the nanoparticle size also became larger and demonstrated a wide/dual-distribution pattern, as shown in Figure 3d–f). Particle size distribution became a dual-modal pattern when the pH of the nanoparticle solution was adjusted to 6.0 and GSH was added (Figure 3g). These results indicate that ChitoHISss nanoparticles have pH and redox sensitivity.

To investigate the changes in the fluorescence properties, Ce6 was loaded into the ChitoHISss nanoparticles, and then aqueous solution of Ce6-incorporated ChitoHISss nanoparticles was incubated in the various pH solutions or in the presence of GSH, as shown in Figure 4. As shown in Figure 4a, the fluorescence intensity of Ce6-loaded ChitoHISss nanoparticles was gradually increased in acidic pH; i.e., a more acidic pH induced higher fluorescence intensity. The fluorescence intensity of Ce6-loaded ChitoHISss nanoparticles also gradually increased when GSH was added to the aqueous solution of ChitoHISss nanoparticles, as shown in Figure 4b. These results indicate that ChitoHISss nanoparticles might be affected by acidic pH and/or the redox state of the nanoparticle solution. These phenomena led to disintegration or swelling of the nanoparticles in the aqueous solution, and then the particle size distribution or fluorescence properties changed.

Figure 5 shows the drug release behavior of CHitoHISss nanoparticles. As shown in Figure 5a, a higher drug content in the nanoparticles resulted in a slower drug release rate. These results might be due to the fact that hydrophobic drugs can be aggregated by hydrophobic interactions at higher drug contents and then liberated slowly. When the pH of the nanoparticle solution was adjusted to an acidic pH, the drug release rate became significantly faster at an acidic pH, and the more acidic pH resulted in a faster drug release rate, as shown in Figure 5b. These results indicate that the ChitoHISss nanoparticles have pH sensitivity and then easily liberate drugs at an acidic pH because they swell, and particle sizes in an acidic pH become larger than those in a basic pH. To study the redox sensitivity of the nanoparticles, DOX-incorporated ChitoHISss nanoparticles were incubated in the presence of GSH, as shown in Figure 5c. When GSH was added to the nanoparticle solutions, the DOX release rate also significantly increased and the drug release rate gradually increased according to the concentration of GSH, indicating that the ChitoHISss nanoparticles had redox sensitivity and then responded to GSH. When the nanoparticle solution was adjusted to an acidic pH as pH 6.0 and then GSH was added to assess the acidic/redox dual sensitive manner of ChitoHISss nanoparticles, the drug release rate was the fastest in all tested environments. These results indicate that the ChitoHISss nanoparticles have pH and redox dual-sensitive behaviors in an aqueous solution. These properties induced a transition in particle size distribution and DOX release rate.

### 3.3. Anticancer Activity of ChitoHISss Nanoparticles In Vitro

HuCC-T1 human cholangiocarcinoma cells were used to study the anticancer activity of DOX-incorporated ChitoHISss nanoparticles. HuCC-T1 cells were exposed to DOX to make DOX-resistant HuCC-T1 cells, as shown in Figure 6. As shown in Figure 6a, the viability of the HuCC-t1 cells was dose-dependently decreased according to the DOX concentration. However, DOX-resistant HuCC-T1 cells were relatively resistant to DOX concentration, i.e., the viability of DOX-resistant HuCC-T1 cells was higher than 50% at 10 µg/mL DOX, whereas the viability of HuCC-T1 cells was less than 30%. However, DOX-incorporated ChitoHISss nanoparticles similarly suppressed the viability of HuCC-T1 cells and DOX-resistant HuCC-T1 cells, as shown in Figure 6b. These results indicate that DOX has low cytotoxicity against DOX-resistant cancer cells or difficulties in intracellular delivery when cancer cells are exposed to DOX repetitively. In particular, cell viability against HuCC-T1 cells or DOX-resistant HuCC-T1 cells dramatically decreased according to the concentration of DOX-incorporated ChitoHISss nanoparticles (Figure 6b) compared to DOX itself (Figure 6a). These results might be due to the fact that DOX-incorporated ChitoHISss nanoparticles easily enter an intracellular compartment of HuCC-T1 cells or DOX-resistant HuCC-T1 cells and then suppress the viability of cancer cells. Empty nanoparticles had no significant cytotoxicity against HuCC-T1 cells and DOX-resistant HuCC-T1 cells, as shown in Figure 6c.

Table 2 shows the IC_50_ values of DOX and DOX-incorporated ChitoHISss nanoparticles. Compared to HuCC-T1 cells, IC_50_ values of DOX at DOX-resistant HuCC-T1 cells significantly increased to higher than 10 µg/mL, whereas this value of DOX-incorporated ChitoHISss nanoparticles revealed 0.68 ± 0.024 µg/mL in DOX-resistant HuCC-T1 cells. These results indicate that DOX-incorporated ChitoHISss nanoparticles have superior anticancer activity both in HuCC-T1 cells and in DOX-resistant HuCC-T1 cells. Figure 7 supports the results of Figure 6, i.e., that DOX treatment against DOX-resistant HuCC-T1 cells resulted in a significant decrease in red fluorescence intensity compared to that of HuCC-T1 cells. These results indicate that DOX uptake by cancer cells is inhibited at DOX-resistant HuCC-T1 cells and then intracellular delivery of DOX itself is decreased, as shown in Figure 7a,b. These behaviors affected the cell viability curves, as shown in Figure 6a,b. When DOX-incorporated ChitoHISss nanoparticles were treated, the fluorescence intensity of DOX-resistant HuCC-T1 cells was not significantly decreased compared to that of HuCC-T1 cells (Figure 7a,b), indicating that DOX-incorporated ChitoHISss nanoparticles can be delivered to the intracellular compartment of DOX-resistant HuCC-T1 cells and then efficiently suppress cancer cells as well as HuCC-T1 cells.

All results were triplicated and are expressed as average ± S.D.

Since ChitoHISss nanoparticles had pH- and redox-sensitive properties, DOX and DOX-incorporated ChitoHISss nanoparticles were assessed with DOX-resistant HuCC-T1 cells at various pHs and in the presence of GSH. As shown in Figure 8a, the viability of DOX-resistant HuCC-T1 cells gradually decreased according to the acidic pH, i.e., the acidic pH resulted in lower cell viability, whereas DOX treatment did not significantly change cell viability at an acidic pH. Upon treatment of DOX-incorporated ChitoHISss nanoparticles, cell viability was gradually decreased according to the concentration of GSH, whereas DOX treatment did not significantly change cell viability, as shown in Figure 8b. These results indicate that DOX-incorporated ChitoHISss nanoparticles have GSH sensitivity and then respond to redox status in cancer cells. The fluorescence observation of the cells also supports these results, as shown in Figure 9. As shown in Figure 9a, treatment of DOX-incorporated ChitoHISss nanoparticles against DOX-resistant HuCC-T1 cells resulted in an increase in red fluorescence intensity at an acidic pH, indicating that they have superior delivery capacity at an acidic pH. The fluorescence intensity of the cancer cells was also gradually increased according to the GSH concentration, indicating that the delivery capacity of DOX-incorporated ChitoHISss nanoparticles was higher at redox status. Therefore, DOX-incorporated ChitoHISss nanoparticles have acidic pH- and redox-sensitive properties.

### 3.4. Antitumor Activity of ChitoHISss Nanoparticles In Vivo

To evaluate the antitumor activity of DOX-incorporated ChitoHISss nanoparticles, a tumor xenograft model was prepared using DOX-resistant HuCC-T1 cells in a BALb/C nude mouse, as shown in Figure 10. Then, DOX solution or DOX-incorporated ChitoHISss nanoparticle solution was intravenously (i.v.) administered via the tail vein of the mouse. The volume of the tumor xenograft gradually increased with control treatment and empty nanoparticles. Practically, empty nanoparticles did not significantly affect the changes in tumor volume growth, as shown in Figure 10a. However, DOX or DOX-incorporated ChitoHISss nanoparticles efficiently inhibited the growth of tumor volumes. Especially, DOX-incorporated ChitoHISss nanoparticles efficiently inhibited tumor growth more than that of DOX itself, indicating that they have superior antitumor activity in an in vivo tumor xenograft model. Changes in body weight with treatment of empty nanoparticles was not significantly different compared to the control treatment, indicating that empty nanoparticles do not affect the body weight of mice. When the mice were treated with DOX and/or DOX-incorporated ChitoHISss nanoparticles, the body weight of mice was slightly decreased compared to the control treatment. These results might be due to the cytotoxicity of DOX. For biodistribution of nanoparticles in vivo, Ce6-incorporated ChitoHISss nanoparticles were i.v. administered via the tail vein of the mouse, as shown in Figure 10c. As shown in Figure 10c, the fluorescence intensity was strongest in the tumor xenograft (left images of whole body) than any other body site. Fluorescence intensity in tumor tissue was stronger than that of other organs, as shown in Figure 10c. These results indicate that ChitoHISss nanoparticles have superior potential in tumor targeting.

## 4. Discussion

For efficient anticancer drug delivery and targeting of tumors, ChitoHISss conjugates were synthesized and DOX-incorporated ChitoHISss nanoparticles were fabricated by the dialysis method. Hydrophobic drugs can be incorporated into the inner core of nanoparticles through hydrophobic interactions with hydrophobic moiety of polymers [37,38]. In addition, higher drug feeding induced higher drug content in the nanoparticles and slower drug release rates from the nanoparticles [37,39]. That is, the hydrophobic agents aggregated in the core of the nanoparticles and then dissolved or were liberated slowly [37]. Since the HIS moiety of conjugates is a hydrophobic molecule and COS itself is a hydrophilic polymer, ChitoHISss conjugates have an amphiphilic property and can form spherical nano-aggregates by a self-assembling process. Fluorescence excitation spectra of pyrene in the presence of ChitoHISss nanoparticles showed that CAC of ChitoHISss nanoparticles was observed at a very low concentration, as shown in Figure 2c, indicating that ChitoHISss conjugates have the potential to form self-aggregates in aqueous solution. Many reports indicated that amphiphilic polymers or conjugates can form self-aggregates [37,38]. For example, Almeida et al. reported that chitosan/polycaprolactone graft copolymer formed polymeric micelles in an aqueous solution and that these micelles formed spherical nanoparticles at very low concentrations [37]. We also previously reported that chitosan-ursodeoxycholic acid conjugates formed self-aggregates in an aqueous solution at very low concentrations and efficiently delivered anticancer drugs to gastrointestinal cancer cells [38]. In this study, we observed spherical nanoparticles of ChitoHISss conjugates in aqueous solution, as shown in Figure 2a, and their particle size distribution revealed a narrow/mono-modal pattern, as shown in Figure 2b. When the pH of nanoparticle solutions was adjusted to an acidic pH and/or GSH was added, the particle size distribution became wide and multi-modal patterns, as shown in Figure 3. These results indicate that ChitoHISss nanoparticles have an acid pH and redox sensitivity. In our previous reports, nanoparticles fabricated from acetyl-histidine-conjugated chitosan copolymer showed pH-sensitive drug delivery properties, i.e., nanoparticles of acetyl-histidine-conjugated chitosan copolymer were disintegrated or swelled at an acidic pH and then the drug release rate was also accelerated at an acidic pH [39]. In this report, the histidine moiety of nanoparticles contributed to the swelling or disintegration of nanoparticles and then led to changes in particle size distribution from a narrow/monomodal pattern to wide/multimodal patterns. Lee and Jeong also reported that nanoparticles composed of hyaluronic acid/poly(l-histidine) copolymer (HAPHSce6ss) with disulfide linkages revealed acid pH- and GSH-sensitive changes in particle sizes, i.e., the average particle size and drug release of HAPHSce6ss nanoparticles increased according to the acidity or GSH concentration of aqueous solutions [40]. In this report, the histidine segment contributed to changes in average particle sizes at an acidic pH. In our results, acid pH and GSH addition led to an increase in particle size and fluorescence intensity of ChitoHISss nanoparticles, as shown in Figure 2, Figure 3 and Figure 4. These results indicate that ChitoHISss nanoparticles possess pH and redox sensitivity.

The tumor microenvironment is known to have an abnormal physiological state, and these tumor characteristics are distinguished from normal tissues [41,42]. They are frequently associated with enhanced metabolism, increased growth rate of tumor cells, overexpression of receptors, acidic pH, and increased redox potential [43,44]. These abnormalities are also associated with the multi-drug resistant (MDR) of tumors [45,46]. Correia and Bissell stated that the abnormality of the tumor microenvironment is one of the dominant factors for MDR of tumors [45]. Wu et al. proposed that adaptive mechanisms of tumor resistance are closely connected to the TME rather than depending on non-cell-autonomous changes in response to clinical treatment [46]. Paradoxically, the abnormal status of the tumor microenvironment has been considered for targeting issues and led the development of various drug-targeting strategies for the conquest of MDR of tumors [28,47]. An increase in lactic acid production, which is a waste product of the tumor metabolic process, is known to induce acidification of the tumor microenvironment and carcinogenicity of tumors, such as invasion/metastasis, angiogenesis, and drug resistance [48]. Furthermore, an imbalance in redox homeostasis in cancer cells simultaneously increases reactive oxygen species (ROS) and antioxidant molecules such as GSH. These systemic imbalances in the tumor microenvironment induce cancer cells to resist against various anticancer agents through the alteration of the drug metabolism, increase in drug efflux rate, pro-survival pathway activation, and slowdown of the apoptosis process [49]. In particular, an increase in the GSH level in cancer cells induces detoxification of anticancer drugs and reduces drug accumulation in cancer cells [50]. Then, an increase in the GSH level in cancer cells is associated with failure of chemotherapy [51]. From these points of view, we decided to fabricate pH- and redox-sensitive nanoparticles to overcome obstacles of cancer therapy through efficient delivery of DOX. Since MDR of tumors against various anticancer agents and their systemic cytotoxicity have also been discussed in CCA patients, nanomedicine such as ChitoHISss nanoparticles may support solutions to overcome the drawbacks of conventional chemotherapy [20]. To make DOX-resistant cancer cells, HuCC-T1 CCA cells were repeatedly exposed to low concentrations of DOX, and then HuCC-T1 cells became resistant to DOX, as shown in Figure 6a. However, DOX-incorporated ChitoHISss nanoparticles showed an almost similar tendency in cell viability, i.e., the viability of DOX-resistant HuCC-T1 cells was dose-dependently inhibited by treatment of DOX-incorporated ChitoHISss nanoparticles, as shown in Figure 6b. Figure 7 supports these results, i.e., a significant decrease in fluorescence intensity in cancer cells was observed with treatment of DOX itself, whereas they were not significantly changed by treatment of DOX-incorporated ChitoHISss nanoparticles. These results indicate that DOX-incorporated ChitoHISss nanoparticles easily enter the intracellular compartment of cancer cells and then kill cancer cells, whereas the uptake of DOX itself is relatively inhibited by cancer cells. These results indicate that ChitoHISss nanoparticles have the potential to overcome MDR of CCA tumors.

The acidic pH of the tumor microenvironment was also considered as a targeting issue for nanomedicine-based drug targeting because nano-dimensional carriers can be designed to be sensitive to acidic pH and then to accelerate the liberation of anticancer drugs in tumor tissue rather than blood neutral/basic pH [52,53]. Hwang et al. reported that pH-sensitive nanoparticles improve intracellular delivery of anticancer drugs and efficiently inhibit the viability of cancer cells at an acidic pH [52]. Garcia et al. also reported that the acidic pH of tumor accelerates DOX release from pH-sensitive liposomes and then efficiently kills breast cancer cells [53]. Our results also show that the DOX release rate from ChitoHISss nanoparticles was accelerated at an acidic pH, such as pH 6.0 and 6.8 (Figure 5b). These properties of ChitoHISss nanoparticles were due to the swelling and/or disintegration of nanoparticles in the acidic pH, as shown in Figure 3 and Figure 4. Then, the DOX release rate from ChitoHISss nanoparticles became higher in an acidic pH than in a neutral or basic pH. ChitoHISss nanoparticles showed increased anticancer activity against DOX-resistant HuCC-T1 cells, as shown in Figure 8a. Palanikumar et al. also reported that DOX-triphenylphosphonium (DOX-TPP) conjugates also showed pH-dependent cytotoxicity against breast cancer cells, i.e., cell viability in a treatment of 2 μg/mL DOX-TPP was less than 20% at pH 6.5, whereas more than 60% of cancer cells were viable at pH 7.4 [54]. We also obtained pH-sensitive delivery of DOX-incorporated ChitoHISss nanoparticles, and the lower the pH the higher the uptake of nanoparticles induced, as shown in Figure 9a. These results indicate that ChitoHISss nanoparticles can be delivered to DOX-resistant HuCC-T1 cells in a pH-sensitive manner.

The higher redox status in the tumor microenvironment is also known to contribute to MDR of tumors [49,50,51]. Paradoxically, an imbalance of redox homeostasis in tumor tissue has also applied to targeting issues in nanomedicine and to overcoming MDR of tumors [55,56,57]. Since disulfide linkage can be degraded in the presence of GSH, nanoparticles with a disulfide linkage have been investigated for cancer cell-specific delivery of anticancer drugs [55,56,57,58]. When nanoparticles are delivered intracellularly in cancer cells, disulfide linkage in the nanoparticle matrix can be degraded, and then this phenomenon accelerates the release of anticancer drugs in the intracellular compartment of cancer cells [55,56]. Li et al. reported that the camptothecin release rate from nanoparticles was significantly increased according to the GSH concentration, i.e., a higher GSH concentration induced a higher drug release rate [56]. Chen et al. also reported that polymer nanoparticles with disulfide linkages liberated DOX in a GSH-specific manner, and nanoparticles resulted in higher tumor inhibition with few side effects [57]. Yoon et al. also reported that nanoparticles with disulfide linkages were able to release anticancer drug in a GSH-specific manner and carry out GSH-dependent intracellular delivery to colon cancer cells [58]. Our results also show that DOX-incorporated ChitoHISss nanoparticles responded to GSH levels in the aqueous system, and the DOX release rate was increased according to the GSH concentration in the solution (Figure 5c). Anticancer activity and intracellular delivery against DOX-resistant HuCC-T1 cells were also improved, as shown in Figure 7, Figure 8 and Figure 9. In our results, ChitoHISss nanoparticles had a similar tendency compared to the results of other reports [55,56,57,58]. In our results, intracellular uptake of free DOX was decreased in DOX-resistant HuCC-T1 cells, whereas ChitoHISss nanoparticles maintained superior intracellular delivery capacity both in normal HuCC-T1 cells and DOX-resistant HuCC-T1 cells (Figure 7). DOX-incorporated ChitoHISss nanoparticles showed superior anticancer activity against DOX-resistant HuCC-T1 cells, whereas free DOX revealed decreased anticancer activity (Figure 6). In addition, DOX-incorporated ChitoHISss nanoparticles efficiently inhibited tumor growth compared to free DOX through tumor-specific delivery capacity (Figure 10). Our results showed that DOX-incorporated ChitoHISss nanoparticles have potential to overcome MDR through acid pH- and redox-sensitive delivery of DOX.

## 5. Conclusions

ChitoHISss conjugates were synthesized by conjugation of HIS to the backbone of COS using disulfide linkages. Spherical nanoparticles of ChitoHISss nanoparticles and DOX incorporation were fabricated with a dialysis procedure. For the fluorescence study, Ce6-incorporated ChitoHISss nanoparticles were also fabricated. DOX-incorporated ChitoHISss nanoparticles have a small diameter less than 200 nm and showed spherical morphology. Particle size distribution of DOX-incorporated ChitoHISss nanoparticles was changed from a monomodal/narrow distribution to wide/multimodal distribution pattern under an acidic pH and in the presence of GSH. These results indicated that ChitoHISss nanoparticles can be disintegrated or swell at an acidic pH and in the presence of GSH. The fluorescence intensity of ChitoHISss nanoparticles was increased according to the acidic pH and GSH addition. The DOX release rate was increased at an acidic pH and in the presence of GSH—i.e., a more acidic pH and higher GSH concentration of nanoparticle solutions induced a higher drug release rate, indicating that an acidic environment and higher redox states accelerate drug release from ChitoHISss nanoparticles. DOX-resistant HuCC-T1 cells were prepared by repetitive exposure of HuCC-T1 cells to DOX. Free DOX showed decreased anticancer activity at DOX-resistant HuCC-T1 cells, whereas DOX-incorporated ChitoHISss nanoparticles still maintained reasonable dose-dependent anticancer activity. These results were due to the improved intracellular delivery of DOX-incorporated ChitoHISss nanoparticles against DOX-resistant HuCC-T1 cells, whereas intracellular uptake of free DOX was significantly decreased. DOX delivery and anticancer activity of ChitoHISss nanoparticles was relatively increased at an acidic pH and in the presence of GSH, indicating that DOX-incorporated ChitoHISss nanoparticles have acidic pH- and redox-sensitive behavior. In an in vivo tumor xenograft model, ChitoHISss nanoparticles were specifically delivered to tumor tissue and DOX-incorporated ChitoHISss nanoparticles efficiently inhibited tumor growth. We suggest that ChitoHISss nanoparticles are a promising candidate for the treatment of CCA.

## Figures and Tables

**Figure 1 materials-15-03795-f001:**
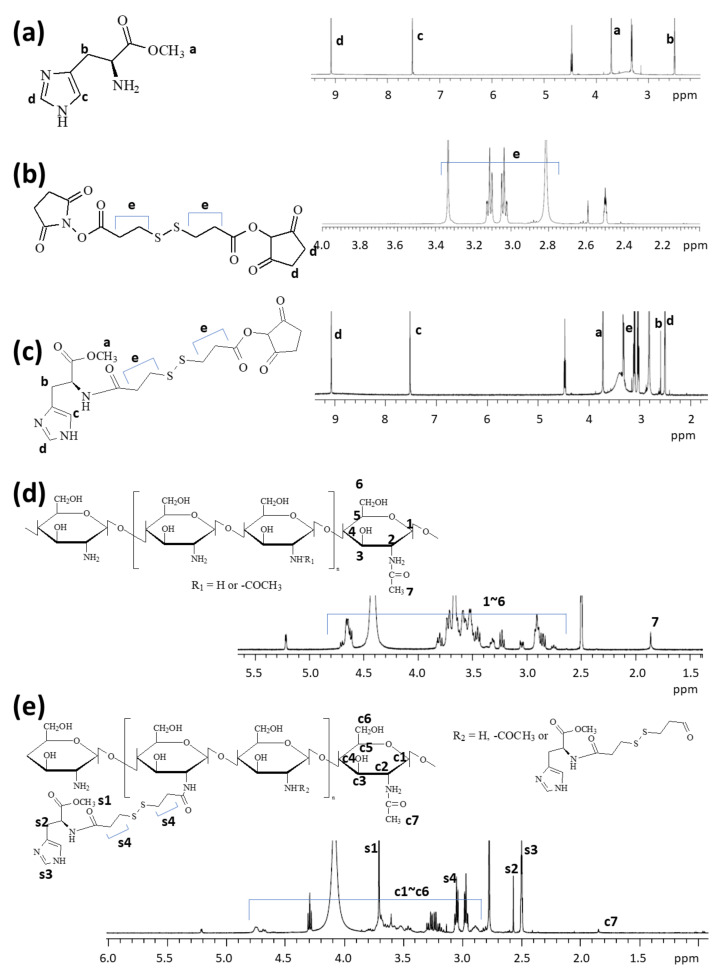
Chemical structure and ^1^H NMR spectra of (**a**) L-histidine methyl ester (HIS), (**b**) dithiodipropionic acid N-hydroxysucinnimide ester (DTP-NHS), (**c**) HIS-DTP-NHS, (**d**) COS, and (**e**) ChitoHISss conjugates.

**Figure 2 materials-15-03795-f002:**
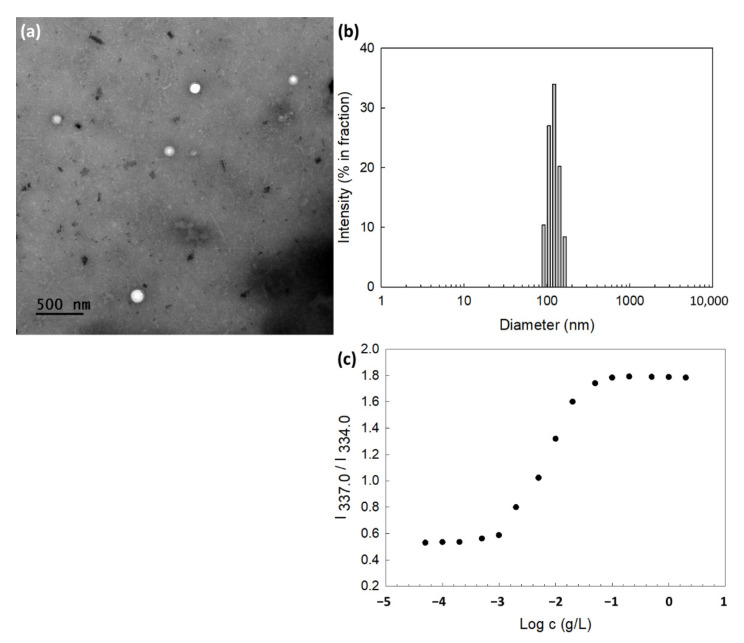
(**a**) Morphological observation and (**b**) typical particle size distribution of ChitoHISss nanoparticles. Particle size distribution is similar to Table 1 (polymer/drug weight ratio = 40/0). Particle size was measured by Zetasizer Nano-ZS^®^ (Malvern, Worcestershire, UK). (**c**) I_337.0_/I_334.0_ intensity ratio plots from pyrene excitation spectra vs. log *c* for ChitoHISss nanoparticles.

**Figure 3 materials-15-03795-f003:**
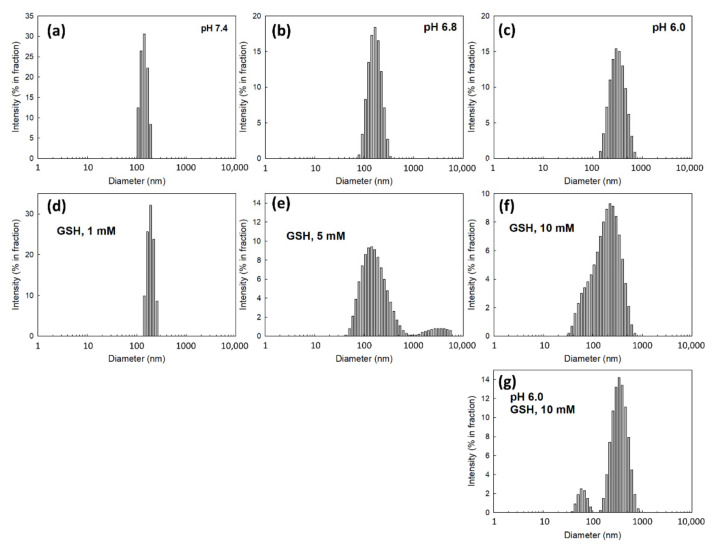
The effect of pH and GSH addition on the changes in particle size distribution of DOX-incorporated ChitoHISss nanoparticles. To assess the effect of pH and GSH on the particle size, 40/5 from Table 1 was used. The effect of pH: (**a**) pH 7.4; (**b**) pH 6.8; (**c**) pH 6.0. The effect of GSH addition. (**d**) GSH, 1 mM; (**e**) GSH, 5 mM; (**f**) GSH, 10 mM. (**g**) pH 6.0 and GSH addition. Nanoparticle solution was incubated at each pH and/or with the addition of GSH for 2 h at 37 °C. Particle size was measured by Zetasizer Nano-ZS^®^ (Malvern, Worcestershire, UK).

**Figure 4 materials-15-03795-f004:**
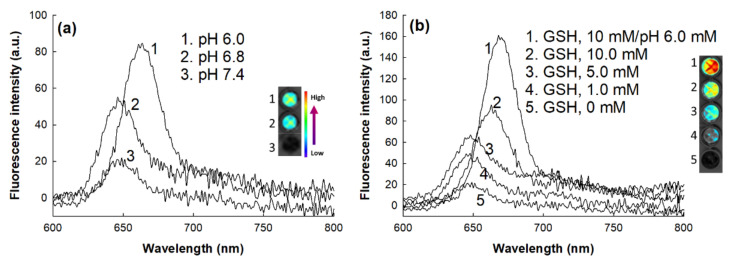
Fluorescence emission spectra of Ce6-loaded ChitoHISss nanoparticles. (**a**) The effect of pH; (**b**) the effect of GSH. Ce6-incorporated ChitoHISss nanoparticles were incubated with various pHs and/or GSH for 4 h. Fluorescence analysis was at least triplicated.

**Figure 5 materials-15-03795-f005:**
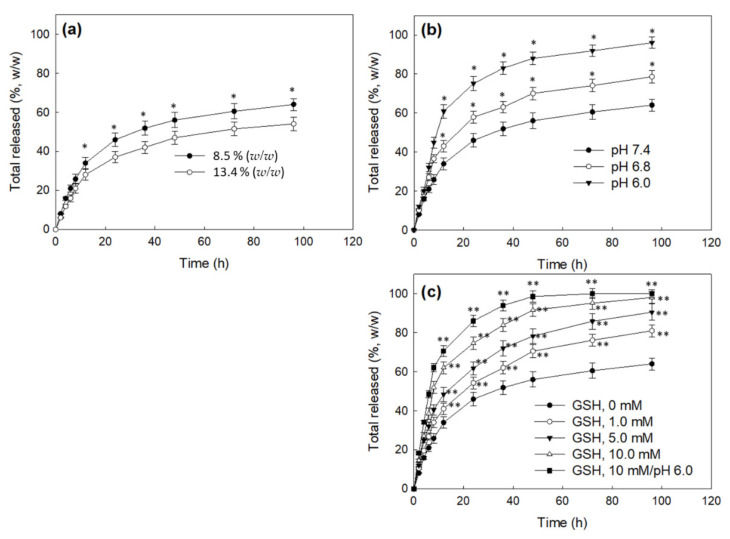
DOX release from ChitoHISss nanoparticles. (**a**) The effect of drug contents; (**b**) the effect of the pH of the nanoparticle solution; (**c**) the effect of GSH addition and acidic pH. All results were triplicated and are expressed as average ± S.D. Statistical analysis: * indicates comparison between 8.5% (*w*/*w*) and 13.4% (*w*/*w*); * also indicates comparison between pH 7.4 and pH 6.8 or pH 6.0.; ** indicates comparison between GSH (0 mM) and GSH 1.0 mM, 5 mM, 10 mM, or GSH 10 mM/pH 6.0. ANOVA followed by Tukey test, *p* < 0.05.

**Figure 6 materials-15-03795-f006:**
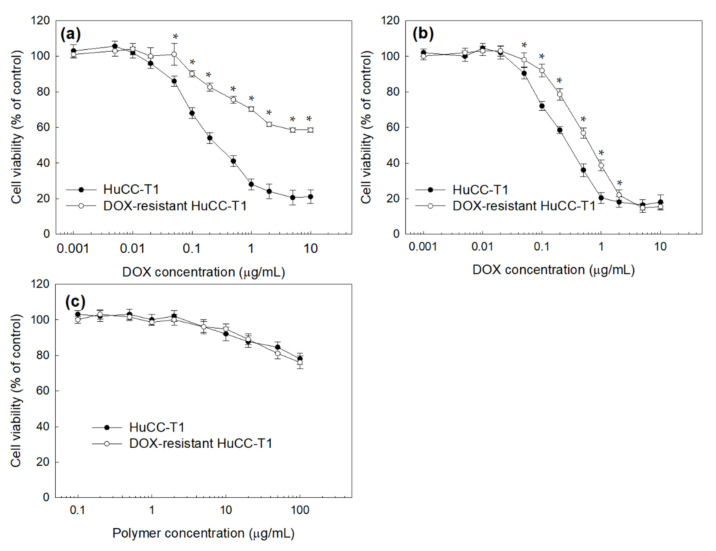
Cell cytotoxicity of DOX or DOX-incorporated ChitoHISss nanoparticles against HuCC-T1 cells and DOX-resistant HuCC-T1 cells. (**a**) DOX; (**b**) DOX-incorporated ChitoHISss nanoparticles; (**c**) empty nanoparticles. Each measurement was the average ± standard deviation (S.D.) from eight wells of 96-well plates. * indicates comparison between HuCC-T1 and DOX-resistant HuCC-T1. ANOVA followed by Tukey test, *p* < 0.05.

**Figure 7 materials-15-03795-f007:**
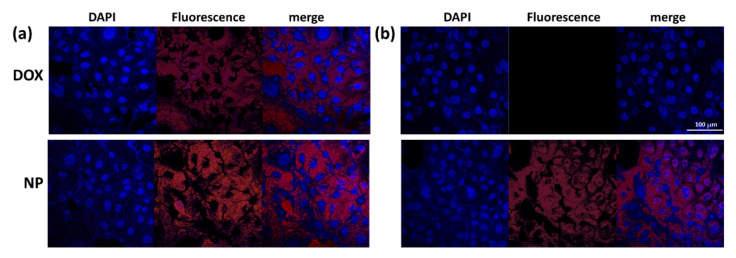
(**a**) The effect of DOX and DOX-incorporated ChitoHISss nanoparticles (NP) on the fluorescence observation of HuCC-T1 cells (**a**) and DOX-resistant HuCC-T1 cells; (**b**) bar = 100 µm.

**Figure 8 materials-15-03795-f008:**
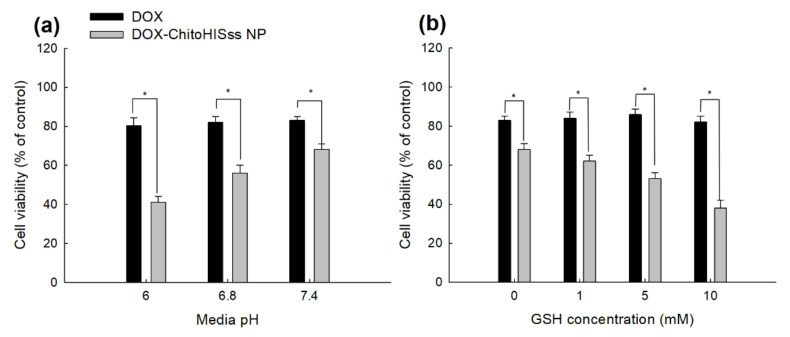
Cell cytotoxicity of DOX and DOX-incorporated ChitoHISss nanoparticles against DOX-resistant HuCC-T1 cells. The effect of media pH (**a**) and the addition of GSH (**b**). For cytotoxicity study, cells were exposed to each pH solution for 6 h and then media were replaced with normal serum-free media. After that, cells were further cultured for 24 h. DOX-ChitoHISss NP: DOX-incorporated ChitoHISss nanoparticles. DOX-concentration: 0.1 µg/mL. Each measurement was average ± standard deviation (S.D.) from eight wells of 96-well plates. * indicates comparison between DOX and DOX-ChitoHISss NP. ANOVA followed by Tukey test, *p* < 0.05.

**Figure 9 materials-15-03795-f009:**
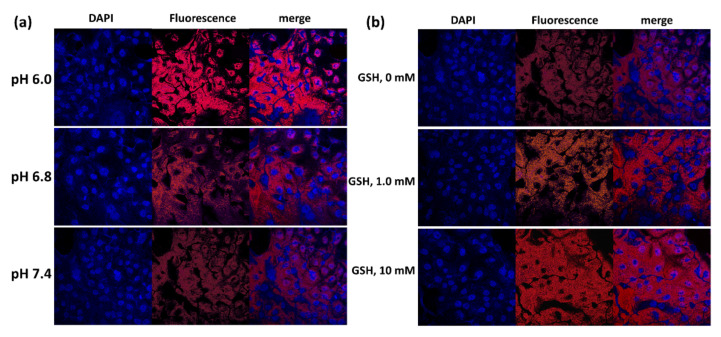
Morphological observation of DOX-resistant HuCC-T1 cells using a confocal microscope. The effect of media pH (**a**) and the addition of GSH (**b**) on the uptake of ChitoHISss nanoparticles in DOX-resistant HuCC-T1 cells. Cells were exposed to Ce6-incorporated ChitoHISss nanoparticles for 90 min.

**Figure 10 materials-15-03795-f010:**
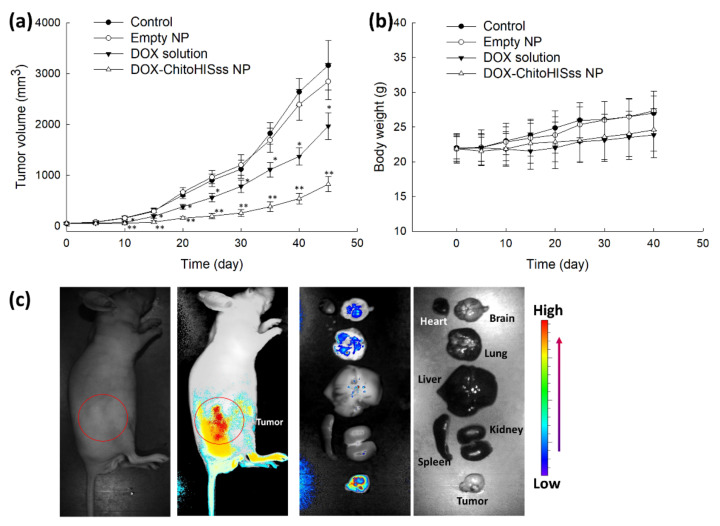
Antitumor activity of DOX-incorporated ChitoHISss nanoparticles. (**a**) Tumor growth; (**b**) body weight changes. The tumor xenograft model of DOX-resistant HuCC-T1 cells was prepared in the back of nude BALb/C mice. DOX-incorporated nanoparticles were i.v. administered through the tail vein (injection volume: 100 µL; dose, 10 mg/kg as a DOX concentration). The treatment dose as a DOX was adjusted to 10 mg/kg. All results are expressed as average ± S.D. from five mice. (**c**) Fluorescence imaging of the tumor-xenograft model of DOX-resistant HuCC-T1 cells. For the fluorescence image, Ce6-incorporated ChitoHISss nanoparticles were i.v. administered via the tail vein of the mouse (10 mg/kg as a Ce6 concentration). The injection volume was 100 µL. One day later, the mice were sacrificed for observation of each organ. Empty NP: empty nanoparticle; DOX-ChitoHISss NP: DOX-incorporated nanoparticles. * indicates comparison between control and DOX solution; ** indicates comparison between control and DOX-ChitoHISss NP. ANOVA followed by Tukey test, *p* < 0.05.

**Table 1 materials-15-03795-t001:** Drug content and particle size of DOX-incorporated ChitoHISss nanoparticles.

Polymer/Drug Weight Ratio (mg/mg)	Drug Content (%, *w*/*w*)		Loading Efficiency (%, *w*/*w*) ^c^	Particle Size (nm)	Polydispersity
	Theoretical ^a^	Experimental ^b^			
40/0	−	−	−	120.5 ± 20.89	0.068
40/5	11.1	8.5	74.3	139.6 ± 24.94	0.075
40/10	20	13.4	61.9	160.5 ± 30.23	0.289

^a^ Theoretical content was calculated from polymer/drug weight ratio. ^b^ Experimental content was measured as depicted in the Materials and Methods section. Drug content (%, *w*/*w*) = (drug weight/nanoparticle weight) × 100. ^c^ Loading efficiency = (drug weight in the nanoparticles/feeding weight of drug) × 100.

**Table 2 materials-15-03795-t002:** IC_50_ value of DOX or DOX-incorporated ChitoHISss nanoparticles.

	IC_50_ (µg/mL) ^a^	
	HuCC-T1 Cells	DOX-Resistant HuCC-T1 Cells
DOX	0.28 ± 0.013	>10
DOX-ChitoHISss NP	0.32 ± 0.012	0.68 ± 0.024
Emp NP	>100	>100

^a^ The IC_50_ value was estimated from the results of Figure 5. DOX-ChitoHISss NP: DOX-incorporated ChitoHISss nanoparticles; empty NP: empty nanoparticles.

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
