# Peer review of "pH and Redox-Dual Sensitive Chitosan Nanoparticles Having Methyl Ester and Disulfide Linkages for Drug Targeting against Cholangiocarcinoma Cells"

_materials, 2022, doi:10.3390/ma15113795_

Round 1

Reviewer 1 Report

In this study an interesting topic is presented, the summary can be improved. Too much information is shown and the contribution or novelty of the proposal is not clear. Interesting results are indicated and supported by an adequate experimental development.

An adequate analysis of the state of the art is shown, where the need to study and propose novel strategies for the administration of anticancer agents against CCA is justified. An analysis of the different types of materials used in similar studies is presented. This allows us to identify the novelty and relevance of the study, as well as the research line of this study. In this study propose to synthesize conjugated composite materials, incorporating chitosan nanoparticles. For the investigation of the potential use for drug administration.

The methodology is described in an appropriate and systematic way.

Section 3.1, is adequately described and shows the NMR characterization of the materials formed. The description is timely and adequate. Literature might be adequate to support results.

Section 3.2: TEM was employed to observe nano-aggregates…. What is the relevance of this size of aggregates, with respect to the study, expand analysis in this regard.

 Line 298-205: “As shown in Table 1, the higher the…….” Why, explain with adequate scientific support what is mentioned here.

Particle size seems to play a very important role, expanding analysis and discussion with adequate scientific support.

Line 349-350: “These results indicated that 349 ChitoHISss nanoparticles can be affected by acidic pH and/or redox state of solution”, what meaning does it have... expand discussion and analysis in this regard.

Figure 5, describe trends or behavior.

Line 417-418: “These results indicated that DOX-incorporated ChitoHISss nanoparticles have GSH-sensitivity and then respond to redox status in biological environment”, because this behavior occurs, expand analysis in this regard, deepen the results obtained.

Author Response

In this study an interesting topic is presented, the summary can be improved. Too much information is shown and the contribution or novelty of the proposal is not clear. Interesting results are indicated and supported by an adequate experimental development.

Answer) Thanks for your comment. We revised the manuscript according to your comment. Practically, we clarified the purpose of this study in the abstract and introduction section.

In Abstract section

The aim of this study is to prepare pH- and redox-sensitive nanoparticles for doxorubicin (DOX) delivery against DOX-resistant HuCC-T1 human cholangiocarcinoma (CCA) cells. For this purpose, L-histidine methyl ester (HIS) was attached to chitosan oligosaccharide (COS) via dithiodipropionic acid (Abbreviated as ChitoHISss). DOX-incorporated nanoparticles of ChitoHISss conjugates were fabricated by dialysis procedure. DOX-resistant HuCC-T1 cells were prepared by repetitive exposure of HuCC-T1 cells to DOX. ChitoHISss nanoparticles have small diameter less than 200 nm and showed spherical morphology. The acid pH and glutathione (GSH) induced changes of size distribution pattern of ChitoHISss nanoparticles from narrow/monomodal distribution pattern to wide/multimodal pattern and increased fluorescence intensity of nanoparticle solution. These results indicated that physicochemical properties of nanoparticles can be changed and/or disintegrated at acidic pH or redox state. The more the acidic pH or the higher the GSH concentration induced the higher drug release rate, indicating that acidic environment or higher redox states accelerated drug release from ChitoHISss nanoparticles. While Free DOX showed decreased anticancer activity at DOX-resistant HuCC-T1 cells, DOX-incorporated ChitoHISss nanoparticles showed dose-dependent anticancer activity. Intracellular delivery of DOX-incorporated ChitoHISss nanoparticles was relatively increased at acidic pH and in the presence of GSH, indicating that DOX-incorporated ChitoHISss nanoparticles have superior acidic pH- and redox-sensitive behavior. In vivo tumor xenograft model, ChitoHISss nanoparticles were specifically delivered to tumor tissue and DOX-incorporated ChitoHISss nanoparticles efficiently inhibited tumor growth. We suggest that ChitoHISss nanoparticles are promising candidate for treatment of CCA.

In Introduction section

The aim of this study is to prepare pH- and redox-sensitive nanoparticles for doxorubicin (DOX) delivery against DOX-resistant HuCC-T1 human cholangiocarcinoma (CCA) cells. For this purpose, L-histidine methyl ester (HIS) was attached to chitosan oligosaccharide (COS) via dithiodipropionic acid (Abbreviated as ChitoHISss).

An adequate analysis of the state of the art is shown, where the need to study and propose novel strategies for the administration of anticancer agents against CCA is justified. An analysis of the different types of materials used in similar studies is presented. This allows us to identify the novelty and relevance of the study, as well as the research line of this study. In this study propose to synthesize conjugated composite materials, incorporating chitosan nanoparticles. For the investigation of the potential use for drug administration.

Answer) Thanks for your comment. According to your comment, we fully revised the manuscript and corrected the English expression.

The methodology is described in an appropriate and systematic way.

Answer) Thanks for your comment. We revised the manuscript according to your comment and the methodology was corrected.

Section 3.1, is adequately described and shows the NMR characterization of the materials formed. The description is timely and adequate. Literature might be adequate to support results.

Answer) Thanks for your comment. According to your comment, we rewritten the manuscript in section 3.1 and literature was added/discussed in the Discussion section.

To synthesize ChitoHISss copolymer, HIS was conjugated with amine groups of COS to endow pH-sensitivity and disulfide linkage was introduced between HIS and COS to endow redox sensitivity. Figure 1 shows the synthesis scheme of ChitoHISss conjugates. As shown in Figure 1(a) and (b), specific peaks of HIS was confirmed at 2 ~ 10 ppm and peaks of ethyl protons of DTP-NHS were confirmed at 2.8 ~ 3.4 ppm. Amine group of HIS was conjugated with the one end NHS group of DTP-NHS to produce HIS-DTP conjugates as shown in Figure 1(c). Specific peaks of HIS and DTP-NHS were confirmed between 2 and 10 ppm, indicating that HIS and TDP- conjugates successfully conjugated. HIS-DTP conjugates were attached again with amine group of COS to make ChitoHISss conjugates as shown in Figure 1(e). 1H NMR spectra of COS was shown in Figure 1(d). As shown in Figure 1(d), specific proton peaks of glucosamine was confirmed between 2.5 and 5.0 ppm. Acetyl group of COS was also confirmed at 1.7 ppm. As shown in Figure 1(e), ChitoHISss conjugates showed specific peaks of COS, HIS and DTP at 1.5 ~5.0 ppm, indicating that HIS-DTP conjugates were successfully conjugated with COS. Yield of final product was estimated by weight measurement and yield was approximately 93.2 % (w/w).

Section 3.2: TEM was employed to observe nano-aggregates…. What is the relevance of this size of aggregates, with respect to the study, expand analysis in this regard.

Answer) Thanks for your comment. Practically, in this study, nano-aggregates means that CHitoHISss conjugates formed nano-sized vehicles as a nanoparticles or nano-aggregates. Anyway, to support this phenomena, we added fluorescence study to clarify aggregation properties of ChitoHISss conjugates in the aqueous solution using pyrene.

Experimental section

To study nano-aggregation behavior of ChitoHISss conjugates, critical aggregation concentration was (CAC) was measured with fluorescence spectrophotometer (Shimadzu RF-5301PC spectrofluorophometer, Kyoto, Japan) using pyrene. 100 μl of pyrene solution in acetone was pipetted into a vial, acetone was evaporated in room temperature, and then 10 ml aqueous nanoparticle solution was poured into vial (Final concentration of pyrene: 6.0×10-7 M). These solutions were equilibrated at 65 oC for 3 hrs following with cooling at room temperature for 2 h. Fluorescence properties of these solutions were measured at 300 nm ~ 350 nm of emission wavelength (excitation wavelength was 390 nm; Excitation and emission band withs, 1.5 nm and 1.5 nm).

Result section

Especially, ChitoHISss has an amphiphilic property and they are able to be aggregated by themselves in an aqueous solution. Critical aggregation concentration (CAC) was evaluated to define their nano-aggregation properties as shown in Figure 2(c). Partition of pyrene (6.0 ×10-7 M) into core of the nanoparticles were assessed as a fluorescence excitation spectra and then red shift of pyrene was observed according to the increase of nanoparticle concentration as shown in Figure 2(c). The (0,0) bands in the excitation spectra of pyrene were compared in the intensity ratio I337.0/I334.0 as shown in Figure 2(c). At fluorescence excitation spectra, cross-over region was observed between flat region and sigmoidal region as shown in Figure 2(c). This region was indicated as a CAC value and the CAC value was approximately 0.0029 g/L.

Figure 2. Morphological observation (a) and typical particle size distribution (b) of ChitoHISss nanoparticles. Particle size distribution is similar to Table 1 (polymer/drug weight ratio = 40/0). (c) I337.0/I334.0 intensity ratio plots from pyrene excitation spectra vs. log c for ChitoHISss nanoparticles.

 Line 298-205: “As shown in Table 1, the higher the…….” Why, explain with adequate scientific support what is mentioned here.

Answer) Thanks for your comment. According to your comment, we revised the manuscript and the explanation for this results was added to results and discussion section.

In Result section

As shown in Table 1, the higher the drug feeding weight induced the higher drug contents, indicating that DOX can be loaded into the nanoparticles through hydrophobic interaction with lipophilic segment (HIS) of ChitoHISss. The higher the drug contents induced the larger particle size as shown in Table 1.

In Discussion section

For efficient anticancer drug delivery and targeting of tumor, ChitoHISss conjugates were synthesized and DOX-incorporated ChitoHISss nanoparticles were fabricated by dialysis method. Hydrophobic drug can be incorporated into the innercore of the nanoparticles through hydrophobic interactions with hydrophobic moiety of polymers [37,38]. Also, the higher the drug feeding induced the higher drug contents in the nanoparticles and the slower drug release rates from the nanoparticles [37,39]. That is, the hydrophobic agents aggregated in the core of the nanoparticles and then dissolved or liberated slowly [37]. Since HIS moiety of conjugates is a hydrophobic molecule and COS itself is a hydrophilic polymer, ChitoHISss conjugates has an amphiphilic property and then they can form spherical nano-aggregates by self-assembling process. Fluorescence excitation spectra of pyrene in the presence of ChitoHISss nanoparticles showed that CAC of ChitoHISss nanoparticles was observed at very low concentration as shown in Figure 2(c), indicating that ChitoHISss conjugates have a potential to form self-aggregates in aqueous solution. Many reports reported that amphiphilic polymers or conjugates can form self-aggregates [37,38]. For example, Almeida et al. reported that chitosan/polycaprolactone graft copolymer formed polymeric micelles in an aqueous solution and these micelles formed spherical nanoparticles at very low concentrations [37]. We also previously reported that chitosan-ursodeoxycholic acid conjugates formed self-aggregates in an aqueous solution at very low concentrations and they efficiently delivered anticancer drugs to gastrointestinal cancer cells [38].

  1. Gref, R.; Minamitake, Y.; Peracchia, M.T.; Trubetskoy, V.; Torchilin, V.; Langer, R. Biodegradable long-circulating polymeric nanospheres. Science. 1994, 263, 1600-1603.
  2. Lee, H.M.; Jeong, Y.I.; Kim, D.H.; Kwak, T.W.; Chung, C.W.; Kim, C.H.; Kang, D.H. Ursodeoxycholic acid-conjugated chitosan for photodynamic treatment of HuCC-T1 human cholangiocarcinoma cells. Int. J. Pharm. 2013, 454, 74-81.
  3. Lee, H.L.; Hwang, S.C.; Nah, J.W.; Kim, J.; Cha, B.; Kang, D.H.; Jeong, Y.I. Redox- and pH-responsive nanoparticles release piperlongumine in a stimuli-sensitive manner to inhibit pulmonary metastasis of colorectal carcinoma cells. J. Pharm. Sci. 2018, 107, 2702-2712. 

Particle size seems to play a very important role, expanding analysis and discussion with adequate scientific support.

Answer) Thanks for your comment. According to your comment, we expand the discussion in the discussion section and then revised the manuscript and added references.

we observed spherical nanoparticles of ChitoHISss conjugates in aqueous solution as shown in Figure 2(a) and their particle size distribution revealed narrow/mono-modal pattern as shown in Figure 2(b). When pH of nanoparticle solutions was adjusted to acidic pH and/or GSH was added, particle size distribution became wide and multi-modal patterns as shown in Figure 3. These results indicated that ChitoHISss nanoparticles have an acid pH- and redox-sensitivity. At our previous reports, nanoparticles fabricated from acetyl-histidine-conjugated chitosan copolymer showed pH-sensitive drug delivery properties, i.e. nanoparticles of acetyl-histidine-conjugated chitosan copolymer was disintegrated or swelled at acidic pH and then drug release rate was also accelerated at acidic pH [39]. In this report, histidine moiety of nanoparticles contributed to the swelling or disintegration of nanoparticles and then led changes of particle size distribution from narrow/monomodal pattern to wide/multimodal patterns. Lee and Jeong also reported that nanoparticles composed of hyaluronic acid/poly(l-histidine) copolymer (HAPHSce6ss) having disulfide linkages revealed acid pH- and GSH-sensitive changes in particle sizes, i.e. average particle size and drug release of HAPHSce6ss nanoparticles was increased according to the acidity or GSH concentration of aqueoussolutions [40]. In this report, histidine segment contributed changes of average particle sizes at acidic pH. In our results, acid pH and GSH addition led increase of particle size and fluorescence intensity of ChitoHISss nanoparticles as shown in Figure 2, 3 and 4. These results indicated that ChitoHISss nanoparticles possess pH and redox-sensitivity.

  1. Gref, R.; Minamitake, Y.; Peracchia, M.T.; Trubetskoy, V.; Torchilin, V.; Langer, R. Biodegradable long-circulating polymeric nanospheres. Science. 1994, 263, 1600-1603.
  2. Lee, H.M.; Jeong, Y.I.; Kim, D.H.; Kwak, T.W.; Chung, C.W.; Kim, C.H.; Kang, D.H. Ursodeoxycholic acid-conjugated chitosan for photodynamic treatment of HuCC-T1 human cholangiocarcinoma cells. Int. J. Pharm. 2013, 454, 74-81.
  3. Lee, H.L.; Hwang, S.C.; Nah, J.W.; Kim, J.; Cha, B.; Kang, D.H.; Jeong, Y.I. Redox- and pH-responsive nanoparticles release piperlongumine in a stimuli-sensitive manner to inhibit pulmonary metastasis of colorectal carcinoma cells. J. Pharm. Sci. 2018, 107, 2702-2712. 
  4. Lee, S.J.; Jeong, Y.I. Hybrid nanoparticles based on chlorin e6-conjugated hyaluronic acid/poly(l-histidine) copolymer for theranostic application to tumors. J. Mater. Chem. B. 2018, 6, 2851-2859.

Line 349-350: “These results indicated that 349 ChitoHISss nanoparticles can be affected by acidic pH and/or redox state of solution”, what meaning does it have... expand discussion and analysis in this regard.

Answer) Thanks for your comment. According to your comment, we expand the discussion in the discussion section and then revised the manuscript and added references.

In Results section

These results indicated that ChitoHISss nanoparticles might be affected by acidic pH and/or redox state of nanoparticle solution. These phenomena led disintegration or swelling of nanoparticles in the aqueous solution and then particle size distribution or fluorescence properties was changed.

In discussion section.

we observed spherical nanoparticles of ChitoHISss conjugates in aqueous solution as shown in Figure 2(a) and their particle size distribution revealed narrow/mono-modal pattern as shown in Figure 2(b). When pH of nanoparticle solutions was adjusted to acidic pH and/or GSH was added, particle size distribution became wide and multi-modal patterns as shown in Figure 3. These results indicated that ChitoHISss nanoparticles have an acid pH- and redox-sensitivity. At our previous reports, nanoparticles fabricated from acetyl-histidine-conjugated chitosan copolymer showed pH-sensitive drug delivery properties, i.e. nanoparticles of acetyl-histidine-conjugated chitosan copolymer was disintegrated or swelled at acidic pH and then drug release rate was also accelerated at acidic pH [39]. In this report, histidine moiety of nanoparticles contributed to the swelling or disintegration of nanoparticles and then led changes of particle size distribution from narrow/monomodal pattern to wide/multimodal patterns. Lee and Jeong also reported that nanoparticles composed of hyaluronic acid/poly(l-histidine) copolymer (HAPHSce6ss) having disulfide linkages revealed acid pH- and GSH-sensitive changes in particle sizes, i.e. average particle size and drug release of HAPHSce6ss nanoparticles was increased according to the acidity or GSH concentration of aqueoussolutions [40]. In this report, histidine segment contributed changes of average particle sizes at acidic pH. In our results, acid pH and GSH addition led increase of particle size and fluorescence intensity of ChitoHISss nanoparticles as shown in Figure 2, 3 and 4. These results indicated that ChitoHISss nanoparticles possess pH and redox-sensitivity.

 Acidic pH of tumor microenvironment has also considered as a targeting issue for nanomedicine-based drug targeting because nano-dimensional carriers can be designed to be sensitive to acidic pH and then to accelerate liberation of anticancer drug in tumor tissue rather than blood neutral/basic pH [52,53]. Hwang et al reported that pH-sensitive nanoparticles improve intracellular delivery of anticancer drugs against cancer cells and efficiently inhibited viability of cancer cells at acidic pH rather than neutral pH [52]. Garcia et al. also reported that acidic pH of tumor accelerates DOX release from pH-sensitive liposomes and then efficiently kills breast cancer cells [53].

Figure 5, describe trends or behavior.

Answer) Thanks for your comment. According to your comment, we rewritten the explanation of Figure 5 in the Results section to clarify trens and behavior.

Figure 5 shows the drug release behavior of CHitoHISss nanoparticles. As shown in Figure 5(a), the higher the drug contents in the nanoparticles resulted in the slower the drug release. These results might be due to that hydrophobic drug at higher drug contents can be aggregated in the core of the nanoparticles by hydrophobic interaction and then released slowly. Since HIS moiety in the backbone of chitosan chain has pH-sensitivity, the pH of aqueous nanoparticle solution was adjusted to acidic pH to study the changes of drug release behavior of nanoparticles as shown in Figure 5(b). Drug release rate from nanoparticles was gradually increased according to the acidity of release medium, indicating that ChitoHISss nanoparticles have pH-sensitivity and then drug release behavior can be controlled by pH of the biological environment. Also, these behaviors must be correlated with the swelling and disintegration of nanoparticles in the acidic pH as shown in Figure 3A and Figure 4(a). That is, changes of nanoparticle sizes in the acidic pH was correlated with changes of drug release behavior and then these behaviors accelerated drug release from nanoparticles. The effect of redox-sensitivity of nanoparticles on the drug release was studied as shown in Figure 5(c). As shown in Figure 5(c), DOX-incorporated ChitoHIS nanoparticles were incubated in the presence of GSH. When GSH was added to nanoparticle solutions, DOX release rate was also significantly increased and drug release rate was gradually increased according to the concentration of GSH. These results were due to the changes of particle size distribution of nanoparticles, i.e. the higher the GSH concentration induced swelling or disintegration of nanoparticles (Figure 3B and C, Figure 4(b)) and then these behaviors led increase of drug release rate. These results also indicated that ChitoHISss nanoparticles have redox-sensitivity. When acidity of nanoparticle solution was adjusted to pH 6.0 and GSH was added, drug release rate was fastest at all tested environment, indicating that ChitoHISss nanoparticles showed pH- and redox- dual sensitive behaviors in the physicochemical properties and drug release.

Line 417-418: “These results indicated that DOX-incorporated ChitoHISss nanoparticles have GSH-sensitivity and then respond to redox status in biological environment”, because this behavior occurs, expand analysis in this regard, deepen the results obtained.

Answer) Thanks for your comment. Practically, to analyze redox behavior of nanoparticles, GSH was added in the nanoparticle solution. Then, we analyzed the changes of particle size distribution and the changes of fluorescence properties as shown in Figure 3 and 4. Anyway, we rewritten the results regarding the effect of GSH addition in the results section.

These results might be due to that disulfide linkage (cystamine linkage) between COS backbone and HIS moiety was disconnected and then HIS moiety was separated from ChitoHISss conjugates and, then, these behaviors induced irregular distribution patterns of nanoparticles as shown in Figure 3B(b), (c) and C(a). Also, these results indicated that ChitoHISss nanoparticles have pH and redox-sensitivity.

The effect of redox-sensitivity of nanoparticles on the drug release was studied as shown in Figure 5(c). As shown in Figure 5(c), DOX-incorporated ChitoHIS nanoparticles were incubated in the presence of GSH. When GSH was added to nanoparticle solutions, DOX release rate was also significantly increased and drug release rate was gradually increased according to the concentration of GSH. These results were due to the changes of particle size distribution of nanoparticles, i.e. the higher the GSH concentration induced swelling or disintegration of nanoparticles (Figure 3B and C, Figure 4(b)) and then these behaviors led increase of drug release rate. These results also indicated that ChitoHISss nanoparticles have redox-sensitivity. When acidity of nanoparticle solution was adjusted to pH 6.0 and GSH was added, drug release rate was fastest at all tested environment, indicating that ChitoHISss nanoparticles showed pH- and redox- dual sensitive behaviors in the physicochemical properties and drug release.

Figure 3. The effect of pH and GSH addition on the changes of particle size distribution of DOX-incorporated ChitoHISss nanoparticles. To assess the effect of pH and GSH on the particle size, 40/5 in Table 1 was used. A. The effect of pH. (a) pH 7.4; (b) pH 6.8; (c) pH 6.0. B. GSH addition. (a) GSH, 1 mM; (b) GSH, 5 mM; (c) GSH, 10 mM. C. pH 6.0 and GSH addition. Nanoparticle solution was incubated at each pH and/or addition of GSH for 2 h at 37oC.

Figure 4. Fluorescence emission spectra of Ce6-loaded ChitoHISss nanoparticles. (a) The effect of pH; (b) The effect of GSH. Ce6-incorporated ChitoHISss nanoparticles were incubated with various pHs and/or GSH for 4 h.

Reviewer 2 Report

The work is well presented and objectives are clear.

I suggest two minor modifications :

Abstract is too long. It should be rewritten to contant main results without extensive describtion of methods used. 

The discussion should be improved by compaing this drug delivery system to previously published. Only conventionally chemotherapy is compared with the systhem. However, many publications treat development of drug delivery nanomaterials. It si not clear what are advantages of the presented one.

Author Response

The work is well presented and objectives are clear.

I suggest two minor modifications :

Abstract is too long. It should be rewritten to contant main results without extensive describtion of methods used. 

Answer) Thanks for your comment. According to your comment, we rewritten the abstract and fully revised the manuscript.

Abstract: The aim of this study is to prepare pH- and redox-sensitive nanoparticles for doxorubicin (DOX) delivery against HuCC-T1 human cholangiocarcinoma (CCA) cells. For this purpose, L-histidine methyl ester (HIS) was attached to chitosan oligosaccharide (COS) via dithiodipropionic acid (Abbreviated as ChitoHISss). DOX-incorporated nanoparticles of ChitoHISss conjugates were fabricated by dialysis procedure. DOX-resistant HuCC-T1 cells were prepared by repetitive exposure of HuCC-T1 cells to DOX. ChitoHISss nanoparticles have small diameter less than 200 nm and showed spherical morphology. The acid pH and the addition of glutathione (GSH) induced changes of size distribution of ChitoHISss nanoparticles from monomodal/narrow distribution to wide/multimodal distribution pattern and increased fluorescence intensity of aqueous nanoparticle solution These results indicated that physicochemical properties of nanoparticles can be changed and/or disintegrated at acidic pH or redox state. The more the acidic pH or the higher GSH concentration induced the higher drug release rate, indicating that acidic environment or higher redox states accelerated drug release from ChitoHISss nanoparticles. While Free DOX showed decreased anticancer activity at DOX-resistant HuCC-T1 cells, DOX-incorporated ChitoHISss nanoparticles showed dose-dependent anticancer activity. Intracellular delivery of DOX-incorporated ChitoHISss nanoparticles was relatively increased at acidic pH and in the presence of GSH, indicating that DOX-incorporated ChitoHISss nanoparticles have superior acidic pH- and redox-sensitive behavior. In vivo tumor xenograft model, ChitoHISss nanoparticles were specifically delivered to tumor tissue and DOX-incorporated ChitoHISss nanoparticles efficiently inhibited tumor growth. We suggest that ChitoHISss nanoparticles are promising candidate for treatment of CCA.

The discussion should be improved by compaing this drug delivery system to previously published. Only conventionally chemotherapy is compared with the systhem. However, many publications treat development of drug delivery nanomaterials. It si not clear what are advantages of the presented one.

Answer) Thanks for your comment. According to your comment, we compared our delivery system with other report and rewritten discussion section. Furthermore, our manuscript fully revised according to your comment.

Tumor microenvironment is known to have abnormal physiological state and then these characteristics of tumor are distinguished from normal tissues [37,38]. They are frequently associated with enhanced metabolism, increased growth rate of tumor cells, overexpression of receptors, acidic pH and increased redox potential [39,40]. Then, these abnormalities are also associated with multi-drug resistant (MDR) of tumor [41,42]. Correia and Bissell reviewed that abnormality of tumor microenvironment is one of the dominant factors for MDR of tumor [41]. Wu et al. proposed that adaptive mechanisms of tumor resistance are closely connected with the TME rather than depending on non-cell-autonomous changes in response to clinical treatment [42]. Paradoxically, abnormal status of tumor microenvironment has been considered as a molecular target and has been developed various strategies to overcome MDR [28,43]. Increase of lactic acid production, which is a waste product of tumor metabolic process, is known to induce acidification of the tumor microenvironment and, then, these status increases carcinogenicity of tumor such as invasion/metastasis, angiogenesis and drug resistance [44]. Furthermore, imbalance of redox homeostasis in cancer cells simultaneously increases reactive oxygen species (ROS) and antioxidant molecules such as GSH. These systemic imbalances in tumor microenvironment induce cancer cells to be resisted against various anticancer agents through alteration of drug metabolism, increase of drug efflux rate, pro-survival pathway activation and slowdown of apoptosis process [45]. Especially, increase of GSH level in cancer cells induces detoxification of anticancer drugs and reduces drug accumulation in cancer cells [46]. Then, increase of GSH level in cancer cells is associated with failure of chemotherapy [47]. From these points of view, we decided to fabricate pH- and redox- sensitive nanoparticles to overcome obstacles of cancer therapy through efficient delivery of DOX. Since MDR of tumor against various anticancer agents and their systemic cytotoxicity has also discussed in CCA patients, nanomedicine such as ChitoHISss nanoparticles may supports solution to overcome drawbacks of conventional chemotherapy [20]. To make DOX-resistant cancer cells, HuCC-T1 CCA cells was repeatedly exposed to low concentrations of DOX and then HuCC-T1 cells became resistant to DOX as shown in Figure 6(a). However, DOX-incorporated ChitoHISss nanoparticles showed almost similar tendency in cell viability, i.e. viability of DOX-resistant HuCC-T1 cells were dose-dependently inhibited by treatment of DOX-incorporated ChitoHISss nanoparticles as shown in Figure 6(b). These results indicated that ChitoHISss nanoparticles have potential to overcome MDR of CCA tumor.

Acidic pH of tumor microenvironment has also considered as a targeting issue for nanomedicine-based drug targeting because nano-dimensional carriers can be designed to be sensitive to acidic pH and then to accelerate liberation of anticancer drug in tumor tissue rather than blood neutral/basic pH [48,49]. Hwang et al reported that pH-sensitive nanoparticles improve intracellular delivery of anticancer drugs against cancer cells and efficiently inhibited viability of cancer cells at acidic pH rather than neutral pH [48]. Garcia et al. also reported that acidic pH of tumor accelerates DOX release from pH-sensitive liposomes and then efficiently kills breast cancer cells [49]. Our results also showed that DOX release rate from ChitoHISss nanoparticles was accelerated at acidic pH such as pH 6.0 and 6.8 (Figure 5(b)). These properties of ChitoHISss nanoparticles are due to the swelling and/or disintegration of nanoparticles in the acidic pH as shown in Figure 3 and 4. Then, DOX release rate from ChitoHISss nanoparticles became higher in acidic pH than that in the neural or basic pH. ChitoHISss nanoparticles showed increased anticancer activity against DOX-resistant HuCC-T1 cells as shown in Figure 8(a). Palanikumar et al. also reported that DOX-triphenylphosphonium (DOX-TPP) conjugates also show pH-dependent cytotoxicity against breast cancer cells, i.e. cell viability in treatment of 2 μg/mL DOX-TPP was less than 20 % at pH 6.5 while higher than 60 % of cancer cells were viable at pH 7.4 [50]. We also obtained pH-sensitive delivery of DOX-incorporated ChitoHISss nanoparticles and the lower the pH induced the higher uptake of nanoparticles as shown in Figure 9(a). These results indicated that ChitoHISss nanoparticles can be delivered to DOX-resistant HuCC-T1 cells by pH-sensitive manner.

The higher redox status in tumor microenvironment is also known to contribute for MDR of tumor [45-47]. Paradoxically, imbalance of redox homeostasis in tumor tissue has also applied for targeting issues in nanomedicine and for overcome MDR of tumor [51-53]. Since disulfide linkage can be degraded in the presence of GSH, nanoparticles having disulfide linkage have been investigated for cancer cell-specific delivery of anticancer drugs [51-54]. When nanoparticles were delivered intracellularly in cancer cells, disulfide linkage in the nanoparticle matrix can be degraded and then this phenomenon accelerates release of anticancer drug in the intracellular compartment of cancer cells [51,52]. Li et al reported that camptothecin release rate from nano-produgs was significantly increased according to the GSH concentration, i.e. the higher the GSH concentration induced the higher drug release rate [52]. Chen et al. also reported that polymer nanoparticles having disulfide linkages liberate DOX with GSH-specific manner and then nanoparticles resulted in higher tumor inhibition with low side effects [53]. Yoon et al. also reported that nanoparticles having disulfide linkages are able to release anticancer drug with GSH-specific manner and then GSH-dependent intracellular delivery to colon cancer cells [54]. Our results also showed that DOX-incorporated ChitoHISss nanoparticles responded to GSH level in the aqueous system and then DOX release rate was increased according to the GSH concentration in the solution (Figure 5(c)). Anticancer activity and intracellular delivery against DOX-resistant HuCC-T1 cells were also improved as shown in Figure 7, 8 and 9. In our results, ChitoHISss nanoparticles have similar tendency compared to the results of other reports [51-54]. In our results, intracellular uptake of free DOX was decreased in DOX-resistant HuCC-T1 cells while ChitoHISss nanoparticles maintained superior intracellular delivery capacity both normal HuCC-T1 cells and DOX-resistant HuCC-T1 cells (Figure 7). DOX-incorporated ChitoHISss nanoparticles showed superior anticancer activity against DOX-resistant HuCC-T1 cells while free DOX revealed decreased anticancer activity (Figure 6). Also, DOX-incorporated ChitoHISss nanoparticles efficiently inhibited tumor growth compared to free DOX through tumor-specific delivery capacity (Figure 10). Our results showed that DOX-incorporated ChitoHISss nanoparticles have potential to overcome MDR through acid pH- and redox-sensitive delivery of DOX.

  1. Liu, Y.; Li, Q.; Zhou, L.; Xie, N.; Nice, E.C.; Zhang, H.; Huang, C.; Lei, Y. Cancer drug resistance: redox resetting renders a way. Oncotarget. 2016, 7, 42740-42761.
  2. Welters, M.J.; Fichtinger-Schepman, A.M.; Baan, R.A.; Flens, M.J.; Scheper, R.J.; Braakhuis, B.J. Role of glutathione, glutathione S-transferases and multidrug resistance-related proteins in cisplatin sensitivity of head and neck cancer cell lines. J. Cancer. 1998, 77, 556-561.
  3. Nunes, S.C.; Serpa, J. Glutathione in ovarian cancer: A double-edged sword. J. Mol. Sci. 2018, 19, 1882.
  4. Hwang, J.H.; Choi, C.W.; Kim, H.W.; Kim, D.H.; Kwak, T.W.; Lee, H.M.; Kim, C.H.; Chung, C.W.; Jeong, Y.I.; Kang, D.H. Dextran-b-poly(L-histidine) copolymer nanoparticles for pH-responsive drug delivery to tumor cells. J. Nanomedicine. 2013, 8, 3197-3207
  5. García, M.C.; Calderón-Montaño, J.M.; Rueda, M.; Longhi, M.; Rabasco, A.M.; López-Lázaro, M.; Prieto-Dapena, F.; González-Rodríguez, M.L. pH-temperature dual-sensitive nucleolipid-containing stealth liposomes anchored with PEGylated AuNPs for triggering delivery of doxorubicin. J. Pharm. 2022, 619, 121691.
  6. Palanikumar, L.; Al-Hosani, S.; Kalmouni, M.; Nguyen, V.P.; Ali, L.; Pasricha, R.; Barrera, F.N.; Magzoub, M. pH-responsive high stability polymeric nanoparticles for targeted delivery of anticancer therapeutics. Commun Biol. 2020, 3. 95.
  7. Ling, X.; Tu, J.; Wang, J.; Shajii, A.; Kong, N.; Feng, C.; Zhang, Y.; Yu, M., Xie, T.; Bharwani, Z.; Aljaeid, B.M.; Shi, B.; Tao, W.; Farokhzad, O.C. Glutathione-responsive prodrug nanoparticles for effective drug delivery and cancer therapy. ACS Nano. 2019, 13, 357-370.
  8. Li, W.; Chen Z.; Liu, X.; Lian, M.; Peng, H.; Zhang, C. Design and evaluation of glutathione responsive glycosylated camptothecin nanosupramolecular prodrug. Drug Deliv. 2021, 28, 1903-1914.
  9. Chen, R.; Ma, Z.; Xiang, Z.; Xia, Y.; Shi, Q.; Wong, S.C.; Yin, J. Hydrogen peroxide and glutathione dual redox-responsive nanoparticles for controlled DOX release. Biosci. 2020, 20, e1900331. 
  10. Yoon, H.M.; Kang, M.S.; Choi, G.E.; Kim, Y.J.; Bae, C.H.; Yu, Y.B.; Jeong, Y.I. Stimuli-responsive drug delivery of doxorubicin using magnetic nanoparticle conjugated poly(ethylene glycol)-g-chitosan copolymer. J. Mol. Sci. 2021, 22, 13169. 

Reviewer 3 Report

The authors present the preparation and evaluation of chitosan-based doxorubicin (DOX) delivery system against HuCC-T1 human cholangiocarcinoma (CCA) cells possessing dual pH- and redox- sensitivity. A commercially available chitosan was modified with specifically functionalized L-histidine methyl ester in order to introduce pH- and redox sensitive groups into the product. The functional polymer was characterized and then co-assembled with DOX to obtain drug nanocarriers applying the dialysis method. The drug loaded nanoparticles showed superior effects during the in vitro and in vivo experiments as compared to the free drug. The positive effects of pH and redox-sensitivity of the carriers were also demonstrated. Moreover, it was shown that the DOX-loaded functional nanoparticles are able to overcome the drug resistance of the tumor cells. The presented manuscript is well-written and could be of interest for the researchers working in the field. However, there are some issues to be addressed before publication.

  1. The molar-mass characteristics of the chitosan (average molar mass, dispersity) should be added in the Experimental part.
  2. Apparently, the authors use TEM to estimate the nanoparticles’ sizes and size distribution. However, those sizes depend on the sample preparation and usually are smaller (due to the dehydration) than those obtained in dispersion. Much more reliable results concerning particle sizes and size-distributions can be obtained from dynamic light scattering (DLS) measurements in aqueous dispersion.
  3. In the Experimental part regarding TEM-description it should be added that TEM was used to observe morphology and to estimate particle sizes and size-distributions.
  4. In Figure 2 and 3 captions should be noted that the sizes and size-distributions were obtained by TEM-analysis.
  5. The authors should check carefully and correct Figure 4(b). According to Figure 4(b) the fluorescence intensity is decreasing with the increase of GSH content which is opposite to the statement made in the text.
  6. Other minor points:
  • page 2 (line 52): should be “have” instead of “has”;
  • page 2 (line 77): should be “is” instead of “are”
  • page 2 (line 94): should be “drug” instead of “rug”;
  • page 2 (line 97): should be “Sun” instead of “sun”;
  • page 4 (line 164): should be “were” instead of “was”;
  • page 4 (line 187): should be “were” instead of “was”;
  • page 5 (line 187): should be “2 days” instead of “2 day”.

Author Response

The authors present the preparation and evaluation of chitosan-based doxorubicin (DOX) delivery system against HuCC-T1 human cholangiocarcinoma (CCA) cells possessing dual pH- and redox- sensitivity. A commercially available chitosan was modified with specifically functionalized L-histidine methyl ester in order to introduce pH- and redox sensitive groups into the product. The functional polymer was characterized and then co-assembled with DOX to obtain drug nanocarriers applying the dialysis method. The drug loaded nanoparticles showed superior effects during the in vitro and in vivo experiments as compared to the free drug. The positive effects of pH and redox-sensitivity of the carriers were also demonstrated. Moreover, it was shown that the DOX-loaded functional nanoparticles are able to overcome the drug resistance of the tumor cells. The presented manuscript is well-written and could be of interest for the researchers working in the field. However, there are some issues to be addressed before publication.

  1. The molar-mass characteristics of the chitosan (average molar mass, dispersity) should be added in the Experimental part.

Answer) Thanks for your comment. We practically purchased from TCI chemical, Co. Ltd, Toyo, Japan (Chitosan Oligosaccharides, Rot. No. C2849). At this moment, we are going to analyze molecular weight/properties of chitosan oligosaccharide and we will analyze chitosan oligosaccharide (used in this study) and will report next paper. Thanks again

  1. Apparently, the authors use TEM to estimate the nanoparticles’ sizes and size distribution. However, those sizes depend on the sample preparation and usually are smaller (due to the dehydration) than those obtained in dispersion. Much more reliable results concerning particle sizes and size-distributions can be obtained from dynamic light scattering (DLS) measurements in aqueous dispersion.

Answer) Thanks for your comment. We apologize we omitted the analysis of particle size distribution. Practically, we measured particle size using zetasizer NanoZS and we indicated it in the experimental part.

In Experimental section

2.7. Analysis of Particle size distribution

Zetasizer Nano-ZS® (Malvern, Worcestershire, UK) was employed to measure particle size distribution. The nanoparticle concentration in the distilled water was adjusted to 0.1 % (w/w) and measured at 20 oC.

  1. In the Experimental part regarding TEM-description it should be added that TEM was used to observe morphology and to estimate particle sizes and size-distributions.

Answer) Thanks for your comment. Practically, we measured particle size using zetasizer and we indicated it in the experimental part. Anyway, according to your comment, we also estimated particle size at TEM photo from TEM equipment and also indicated in the photo and then added as supporting materials file. Thanks again.

In Experimental section

2.7. Analysis of Particle size distribution

Zetasizer Nano-ZS® (Malvern, Worcestershire, UK) was employed to measure particle size distribution. The nanoparticle concentration in the distilled water was adjusted to 0.1 % (w/w) and measured at 20 oC.

In Results section

Figure 2. Morphological observation (a) and typical particle size distribution (b) of ChitoHISss nanoparticles. Particle size distribution is similar to Table 1 (polymer/drug weight ratio = 40/0). Particle size was measured by Zetasizer Nano-ZS® (Malvern, Worcestershire, UK). (c) I337.0/I334.0 intensity ratio plots from pyrene excitation spectra vs. log c for ChitoHISss nanoparticles.

Figure 3. The effect of pH and GSH addition on the changes of particle size distribution of DOX-incorporated ChitoHISss nanoparticles. To assess the effect of pH and GSH on the particle size, 40/5 in Table 1 was used. A. The effect of pH. (a) pH 7.4; (b) pH 6.8; (c) pH 6.0. B. GSH addition. (a) GSH, 1 mM; (b) GSH, 5 mM; (c) GSH, 10 mM. C. pH 6.0 and GSH addition. Nanoparticle solution was incubated at each pH and/or addition of GSH for 2 h at 37oC. Particle size was measured by Zetasizer Nano-ZS® (Malvern, Worcestershire, UK).

  1. In Figure 2 and 3 captions should be noted that the sizes and size-distributions were obtained by TEM-analysis.

Answer) Thanks for your comment. Practically, we measured particle size using zetasizer and we indicated it in the experimental part. Anyway, according to your comment, we indicated the size and size-distribution in the Figure caption. Thanks again.

Figure 2. Morphological observation (a) and typical particle size distribution (b) of ChitoHISss nanoparticles. Particle size distribution is similar to Table 1 (polymer/drug weight ratio = 40/0). Particle size was measured by Zetasizer Nano-ZS® (Malvern, Worcestershire, UK). (c) I337.0/I334.0 intensity ratio plots from pyrene excitation spectra vs. log c for ChitoHISss nanoparticles.

Figure 3. The effect of pH and GSH addition on the changes of particle size distribution of DOX-incorporated ChitoHISss nanoparticles. To assess the effect of pH and GSH on the particle size, 40/5 in Table 1 was used. A. The effect of pH. (a) pH 7.4; (b) pH 6.8; (c) pH 6.0. B. GSH addition. (a) GSH, 1 mM; (b) GSH, 5 mM; (c) GSH, 10 mM. C. pH 6.0 and GSH addition. Nanoparticle solution was incubated at each pH and/or addition of GSH for 2 h at 37oC. Particle size was measured by Zetasizer Nano-ZS® (Malvern, Worcestershire, UK).

In supporting materials

Experimental

Analysis of Particle size distribution

Zetasizer Nano-ZS® (Malvern, Worcestershire, UK) was employed to measure particle size distribution. The nanoparticle concentration in the distilled water was adjusted to 0.1 % (w/w) and measured at 20 oC.

Results

  As shown in Figure S1, TEM photo indicated that ChitoHISss nanoparticles have spherical shapes and their particle sizes were less than 200 nm. Average particle size of ChitoHISss nanoparticles was 134.5±18.4 nm. Their average particle sizes were almost similar to the results of the particle size analysis results as shown in Figure 2 and Table 1.

Figure S1. Morphological observation of ChitoHISss nanoparticles and their particle size. Particle size indicated in the photo was estimated by photo of TEM equipment.

  1. The authors should check carefully and correct Figure 4(b). According to Figure 4(b) the fluorescence intensity is decreasing with the increase of GSH content which is opposite to the statement made in the text.

Answer) Thanks for your comment. We apologize Figure 4(b) is our mistake. Fluorescence intensity was increased according to the concentration of GSH. We corrected it. Appreciate to your indication. Thanks again.

  1. Other minor points:
  • page 2 (line 52): should be “have” instead of “has”;
  • page 2 (line 77): should be “is” instead of “are”
  • page 2 (line 94): should be “drug” instead of “rug”;
  • page 2 (line 97): should be “Sun” instead of “sun”;
  • page 4 (line 164): should be “were” instead of “was”;
  • page 4 (line 187): should be “were” instead of “was”;
  • page 5 (line 187): should be “2 days” instead of “2 day”.

Answer) Thanks for your comment. According to your comment, we corrected the mistyping and grammatical errors. Than

Reviewer 4 Report

Journal: Materials

Manuscript ID: materials-1725089

Title: pH and redox-dual sensitive chitosan nanoparticles having L-histidine methyl ester and disulfide linkages for drug targeting against cholangiocarcinoma cells

Authors: Ju-Il Yang, Hye Lim Lee, Je-Jung Yun, Jungsoo Kim, Kyoung-Ha So,
Young-IL Jeong, Dae-Hwan Kang

In this manuscript submitted as Article to Materials, the authors reported very interesting results of a complex, well designed and conducted study on synthesis, characterization, and antitumor (in vitro and in vivo) effects of chitosan nanoparticles functionalized as carriers for doxorubicin. The topic of this manuscript is relevant to the field of the Materials and fits with the scope of this journal. The paper is well organized, and concise. The Abstract is comprehensive. The presentation of methodology is appropriate, and the results are very clear presented using a large number of illustrative figures. The conclusions are based on the obtained results. The authors used a reasonable number of references for preparing the manuscript, many of them recently published, indicating the actuality of the subject. However, before recommending the publication into the Materials journal, a minor (but extensive) revision of the manuscript is required.

Scientific minor points to be solved:

  1. The Introduction section is not very cursive and there are no connections between the different (otherwise important to be presented) ideas.
  2. The Objectives of the study are not clearly stated at the end of the Introduction;
  3. The authors have to seriously improve the Discussion section. At this moment, the Discussion largely contains literature data and some of the results given again. Into the Discussion, the authors have to compare and discuss the various obtained results, to discuss the relevance of their results and to indicate the novelty of their study. They should also try to explain why their nanoparticles could be considered better than nanoparticles made of other materials, and why is important to obtain reduced sizes of tumors after a single treatment (maybe to discuss the possible effects of multiple treatments in order to totally eradicate the tumor). Finally, the authors have to mention limitations of this study (if it is the case) as well as the perspectives opened by their work.
  4. All acronyms have to be explained at first use in the text (The abstract stands separated).
  5. The authors have to provide the number of measurements performed for each test (n) in sections 2.10, 2.11, 2.12, Table 1, Figs. 4,5,6,8,10.
  6. For statistical analysis, the authors have to use ANOVA instead of Student’s test, allowing comparison of data from all analyzed groups. Statistical significance has to be presented for all obtained results that were compared.
  7. The authors should provide the full origin of the reagents, apparatuses and software used (company, city, [state], country).
  8. In section 3.3, the authors have to mention that Dox-ChitoHISss reduced dramatically cell viability.

Other general minor points to be solved:

The most important general point to be solved is related to the English language. An extensive correction of grammar errors is required, especially for Introduction, but many other errors (including typing errors) appear in all the manuscript. Also, many sentences have to be rephrased. In the Materials and Methods and in the Results sections the authors have to use Past Tense. Few examples of problems to be corrected:

- please rephrase sentences (including the use Past Tense) in lines: 24,27,28,41,49,50,52-54,55-56,64,65-66,67,69,70-71,73-74,77,84,92,94,96-97,98-99,100-102,125,140-141,145,151-152,152-153,164,180,187-188,191,197,198,202-203,208,217,218,220,231,257,281,288,330,352-353,357,367,369,377,380,380-381,382-383,384,386,386,390-391,393,395,414,415,418,419,422,425,443,447,448,449,450-451,459,484-485,487,492,537,541,551;

- in lines 57 and 61 please remove coma between subject and predicate;

- [21] should be replaced with [19] in line 75, and [19] should be replaced with [21] in line 76

- in line 87: “known to be acidic less than pH 7.0” is a pleonasm. There is no acidic pH over pH 7.0. Also, please replace “peculiarity” with another, more appropriate word in the same line;

- typing errors: lines 23 (a infrared), 90 (acidic H), 93 (rug), 97 (sun), 245 (2.11), 348 (increased);

- the word “Furthermore” is overused.

Author Response

Journal: Materials

Manuscript ID: materials-1725089

Title: pH and redox-dual sensitive chitosan nanoparticles having L-histidine methyl ester and disulfide linkages for drug targeting against cholangiocarcinoma cells

Authors: Ju-Il Yang, Hye Lim Lee, Je-Jung Yun, Jungsoo Kim, Kyoung-Ha So,
Young-IL Jeong, Dae-Hwan Kang

In this manuscript submitted as Article to Materials, the authors reported very interesting results of a complex, well designed and conducted study on synthesis, characterization, and antitumor (in vitro and in vivo) effects of chitosan nanoparticles functionalized as carriers for doxorubicin. The topic of this manuscript is relevant to the field of the Materials and fits with the scope of this journal. The paper is well organized, and concise. The Abstract is comprehensive. The presentation of methodology is appropriate, and the results are very clear presented using a large number of illustrative figures. The conclusions are based on the obtained results. The authors used a reasonable number of references for preparing the manuscript, many of them recently published, indicating the actuality of the subject. However, before recommending the publication into the Materials journal, a minor (but extensive) revision of the manuscript is required.

Scientific minor points to be solved:

  1. The Introduction section is not very cursive and there are no connections between the different (otherwise important to be presented) ideas.

Answer) Thanks for your comment. According to your comment, we fully revised the manuscript and the introduction part was modified. Thanks again.

Cholangiocarcinoma (CCA), which is a malignant tumor in the epithelium of the biliary tract, is frequently shown in poor prognosis and the incidence rate of CCA is increasing worldwide [1-3]. Since early diagnosis of CCA is difficult and then is frequently diagnosed as an advanced stage, surgical resection, which is a curative option, is practically unable [4,5]. Except for surgical resection, treatment options such as stent displacement, radiotherapy, chemotherapy and immunotherapy have tried to treat CCA in last several decades [6-9]. Among them, chemotherapy has frequently considered to improve survivability and life quality of CCA patients [10-13]. Clinical trials of chemotherapeutic agents including cisplatin, epirubicin, 5-fluorouracil and gemcitabine have tried to manage biliary tract adenocarcinoma with manageable toxicity against patients [10]. Kim et al. also reported that combination of gemcitabine and cisplatin was tolerable for patients having inoperable biliary tract cancer and showed modest response rates [11]. Combination of cisplatin and gemcitabine is believed to be a synergistic candidate for biliary tract cancer compared to single treatment [12]. Wang et al. reported that hepatic arterial infusion of oxaliplatin and 5-fluorouracil has a benefit to control tumor progression, survivability of patients and toxicity for advanced perihilar cholangiocarcinoma (PCC) [13]. Also, it was reported that chemotherapy following with radiation therapy has beneficial effect against unresectable perihilar CCA [14]. However, most of the treatment regimens such as chemotherapy and radiotherapy have no gain in survivability of patients [15,16]. From these points of view, targeted therapy using molecular targeted agents has been tried to improve therapeutic efficacy and survival period of CCA patients [16,17]. Even though molecular targeted agents have suggested as a promising candidate for targeted therapy, efficacy of them still provides insignificant benefit in survivability of CCA patients [16-19]. Multi-drug resistant (MDR) of CCA against conventional chemotherapeutic agents and/or molecular targeted agents is also problematic in improvement of therapeutic responses and patient survivability [19-21]. For example, Chakrabarti et al. reported that drug resistant problem of fibroblast growth factor receptor (FGFR) inhibitors is problematic and has to be solved for future trials [19]. Massa et al. also reported that paclitaxel-incorporated albumin nanoparticles have benefit to overcome MDR and then to delay tumor growth/vasculature [21]. Therefore, novel anticancer agents based on nanoparticles should be developed to overcome MDR of CCA.

Nano-scale based carriers such as liposomes, nanoparticles and polymeric micelles have been extensively investigated for tumor-specific delivery of bioactive agents [22-26]. Nanoparticles are frequently employed to deliver the anticancer drug against solid tumor because they have huge surface area for ease of modification, small diameter to avoid reticuloendothelial system and structural peculiarity to payload hydrophobic drugs [27]. Especially, biochemical and physiological status of tumor microenvironment is quite different compared to normal tissues [28]. Physiological peculiarities of tumor tissues are an acidic pH environment, vascularization, elevated levels of reduction/oxidation (redox) potential, expression of various molecular receptors, changes of perfusion rate, leaky blood vessels and etc. [28-30]. Especially, acid pH of tumor microenvironment has been applied to control drug delivery behavior of nanocarriers in tumor tissues [31]. Du et al. reported that properties of nanoparticles can be changed to adapt to acidic pH of tumor extracellular environment and intracellular environment [32]. They argued that nanoparticles having acid-cleavable group have sensitivity against the acidic pH of tumor microenvironment and then improve drug delivery capacity. Otherwise, it was reported that glutathione (GSH) levels in tumor microenvironment are significantly higher than normal tissues [33]. The elevated levels of GSH in tumor tissues frequently associate with drug resistant problem [34]. Sun et al. reported that DOX release rate from polymeric micelles having disulfide linkages is accelerated in intracellular compartment of tumor cells because intracellular GSH level in tumor cells is extremely higher than extracellular GSH level and disulfide bond is able to be disintegrated by GSH [35]. Park et al. also reported that polymer nanoparticles having disulfide linkages were cleaved by GSH and cancer cell viability was efficiently inhibited through redox-sensitive delivery of anticancer drugs against cancer cells [36].

  1. The Objectives of the study are not clearly stated at the end of the Introduction;

Answer) Thanks for your comment. According to your comment,

In this study, we synthesized chitosan-histidine conjugates using disulfide linkage (ChitoHISss) and fabricated nanoparticles to overcome MDR of CCA cells. Also, doxorubicin (DOX)-incorporated ChitoHISss nanoparticles fabricated for pH- and redox-sensitive delivery of DOX against HuCC-T1 human cholangiocarcinoma cells. L-histidine methyl ester and cystamine was employed to endow pH and redox-sensitivity to chitosan nanoparticles since histidine has an acidic-sensitivity and cystamine can be cleaved by GSH. DOX-resistant CCA cells were prepared for investigation of drug delivery potential of ChitoHISss nanoparticles.

  1. The authors have to seriously improve the Discussion section. At this moment, the Discussion largely contains literature data and some of the results given again. Into the Discussion, the authors have to compare and discuss the various obtained results, to discuss the relevance of their results and to indicate the novelty of their study. They should also try to explain why their nanoparticles could be considered better than nanoparticles made of other materials, and why is important to obtain reduced sizes of tumors after a single treatment (maybe to discuss the possible effects of multiple treatments in order to totally eradicate the tumor). Finally, the authors have to mention limitations of this study (if it is the case) as well as the perspectives opened by their work.

Answer) Thanks for your comment. According to your comment, we discussed again in the Discussion section and compared with other reports. References also added to the Reference section.

For efficient anticancer drug delivery and targeting of tumor, ChitoHISss conjugates were synthesized and DOX-incorporated ChitoHISss nanoparticles were fabricated by dialysis method. Hydrophobic drug can be incorporated into the innercore of the nanoparticles through hydrophobic interactions with hydrophobic moiety of polymers [37,38]. Also, the higher the drug feeding induced the higher drug contents in the nanoparticles and the slower drug release rates from the nanoparticles [37,39]. That is, the hydrophobic agents aggregated in the core of the nanoparticles and then dissolved or liberated slowly [37]. Since HIS moiety of conjugates is a hydrophobic molecule and COS itself is a hydrophilic polymer, ChitoHISss conjugates has an amphiphilic property and then they can form spherical nano-aggregates by self-assembling process. Fluorescence excitation spectra of pyrene in the presence of ChitoHISss nanoparticles showed that CAC of ChitoHISss nanoparticles was observed at very low concentration as shown in Figure 2(c), indicating that ChitoHISss conjugates have a potential to form self-aggregates in aqueous solution. Many reports reported that amphiphilic polymers or conjugates can form self-aggregates [37,38]. For example, Almeida et al. reported that chitosan/polycaprolactone graft copolymer formed polymeric micelles in an aqueous solution and these micelles formed spherical nanoparticles at very low concentrations [37]. We also previously reported that chitosan-ursodeoxycholic acid conjugates formed self-aggregates in an aqueous solution at very low concentrations and they efficiently delivered anticancer drugs to gastrointestinal cancer cells [38]. In this study, we observed spherical nanoparticles of ChitoHISss conjugates in aqueous solution as shown in Figure 2(a) and their particle size distribution revealed narrow/mono-modal pattern as shown in Figure 2(b). When pH of nanoparticle solutions was adjusted to acidic pH and/or GSH was added, particle size distribution became wide and multi-modal patterns as shown in Figure 3. These results indicated that ChitoHISss nanoparticles have an acid pH- and redox-sensitivity. At our previous reports, nanoparticles fabricated from acetyl-histidine-conjugated chitosan copolymer showed pH-sensitive drug delivery properties, i.e. nanoparticles of acetyl-histidine-conjugated chitosan copolymer was disintegrated or swelled at acidic pH and then drug release rate was also accelerated at acidic pH [39]. In this report, histidine moiety of nanoparticles contributed to the swelling or disintegration of nanoparticles and then led changes of particle size distribution from narrow/monomodal pattern to wide/multimodal patterns. Lee and Jeong also reported that nanoparticles composed of hyaluronic acid/poly(l-histidine) copolymer (HAPHSce6ss) having disulfide linkages revealed acid pH- and GSH-sensitive changes in particle sizes, i.e. average particle size and drug release of HAPHSce6ss nanoparticles was increased according to the acidity or GSH concentration of aqueous solutions [40]. In this report, histidine segment contributed changes of average particle sizes at acidic pH. In our results, acid pH and GSH addition led increase of particle size and fluorescence intensity of ChitoHISss nanoparticles as shown in Figure 2, 3 and 4. These results indicated that ChitoHISss nanoparticles possess pH and redox-sensitivity.

Tumor microenvironment is known to have abnormal physiological state and then these characteristics of tumor are distinguished from normal tissues [41,42]. They are frequently associated with enhanced metabolism, increased growth rate of tumor cells, overexpression of receptors, acidic pH and increased redox potential [43,44]. Then, these abnormalities are also associated with multi-drug resistant (MDR) of tumor [45,46]. Correia and Bissell reviewed that abnormality of tumor microenvironment is one of the dominant factors for MDR of tumor [45]. Wu et al. proposed that adaptive mechanisms of tumor resistance are closely connected with the TME rather than depending on non-cell-autonomous changes in response to clinical treatment [46]. Paradoxically, abnormal status of tumor microenvironment has considered for targeting issues and led development of various drug targeting strategies for conquest of MDR of tumor [28,47]. Increase of lactic acid production, which is a waste product of tumor metabolic process, is known to induce acidification of the tumor microenvironment and carcinogenicity of tumor such as invasion/metastasis, angiogenesis and drug resistance [48]. Furthermore, imbalance of redox homeostasis in cancer cells simultaneously increases reactive oxygen species (ROS) and antioxidant molecules such as GSH. These systemic imbalances in tumor microenvironment induce cancer cells to be resisted against various anticancer agents through alteration of drug metabolism, increase of drug efflux rate, pro-survival pathway activation and slowdown of apoptosis process [49]. Especially, increase of GSH level in cancer cells induces detoxification of anticancer drugs and reduces drug accumulation in cancer cells [50]. Then, increase of GSH level in cancer cells is associated with failure of chemotherapy [51]. From these points of view, we decided to fabricate pH- and redox- sensitive nanoparticles to overcome obstacles of cancer therapy through efficient delivery of DOX. Since MDR of tumor against various anticancer agents and their systemic cytotoxicity has also discussed in CCA patients, nanomedicine such as ChitoHISss nanoparticles may supports solution to overcome drawbacks of conventional chemotherapy [20]. To make DOX-resistant cancer cells, HuCC-T1 CCA cells was repeatedly exposed to low concentrations of DOX and then HuCC-T1 cells became resistant to DOX as shown in Figure 6(a). However, DOX-incorporated ChitoHISss nanoparticles showed almost similar tendency in cell viability, i.e. viability of DOX-resistant HuCC-T1 cells were dose-dependently inhibited by treatment of DOX-incorporated ChitoHISss nanoparticles as shown in Figure 6(b). Figure 7 supported these results, i.e. significant decrease of fluorescence intensity in cancer cells was observed at treatment of DOX itself while those were not significantly changed by treatment of DOX-incorporated ChitoHISss nanoparticles. These results indicated that DOX-incorporated ChitoHISss nanoparticles are easily entered into the intracellular compartment of cancer cells and then kill cancer cells while uptake of DOX itself was relatively inhibited by cancer cells. These results indicated that ChitoHISss nanoparticles have potential to overcome MDR of CCA tumor.

Acidic pH of tumor microenvironment has also considered as a targeting issue for nanomedicine-based drug targeting because nano-dimensional carriers can be designed to be sensitive to acidic pH and then to accelerate liberation of anticancer drug in tumor tissue rather than blood neutral/basic pH [52,53]. Hwang et al. reported that pH-sensitive nanoparticles improve intracellular delivery of anticancer drugs and efficiently inhibit viability of cancer cells at acidic pH rather [52]. Garcia et al. also reported that acidic pH of tumor accelerates DOX release from pH-sensitive liposomes and then efficiently kills breast cancer cells [53]. Our results also showed that DOX release rate from ChitoHISss nanoparticles was accelerated at acidic pH such as pH 6.0 and 6.8 (Figure 5(b)). These properties of ChitoHISss nanoparticles are due to the swelling and/or disintegration of nanoparticles in the acidic pH as shown in Figure 3 and 4. Then, DOX release rate from ChitoHISss nanoparticles became higher in acidic pH than that in the neural or basic pH. ChitoHISss nanoparticles showed increased anticancer activity against DOX-resistant HuCC-T1 cells as shown in Figure 8(a). Palanikumar et al. also reported that DOX-triphenylphosphonium (DOX-TPP) conjugates also show pH-dependent cytotoxicity against breast cancer cells, i.e. cell viability in treatment of 2 μg/mL DOX-TPP was less than 20 % at pH 6.5 while higher than 60 % of cancer cells were viable at pH 7.4 [54]. We also obtained pH-sensitive delivery of DOX-incorporated ChitoHISss nanoparticles and the lower the pH induced the higher uptake of nanoparticles as shown in Figure 9(a). These results indicated that ChitoHISss nanoparticles can be delivered to DOX-resistant HuCC-T1 cells by pH-sensitive manner.

The higher redox status in tumor microenvironment is also known to contribute for MDR of tumor [49-51]. Paradoxically, imbalance of redox homeostasis in tumor tissue has also applied for targeting issues in nanomedicine and for overcome MDR of tumor [55-57]. Since disulfide linkage can be degraded in the presence of GSH, nanoparticles having disulfide linkage have been investigated for cancer cell-specific delivery of anticancer drugs [55-58]. When nanoparticles were delivered intracellularly in cancer cells, disulfide linkage in the nanoparticle matrix can be degraded and then this phenomenon accelerates release of anticancer drug in the intracellular compartment of cancer cells [55,56]. Li et al reported that camptothecin release rate from nano-produgs was significantly increased according to the GSH concentration, i.e. the higher the GSH concentration induced the higher drug release rate [56]. Chen et al. also reported that polymer nanoparticles having disulfide linkages liberate DOX with GSH-specific manner and then nanoparticles resulted in higher tumor inhibition with low side effects [57]. Yoon et al. also reported that nanoparticles having disulfide linkages are able to release anticancer drug with GSH-specific manner and then GSH-dependent intracellular delivery to colon cancer cells [58]. Our results also showed that DOX-incorporated ChitoHISss nanoparticles responded to GSH level in the aqueous system and then DOX release rate was increased according to the GSH concentration in the solution (Figure 5(c)). Anticancer activity and intracellular delivery against DOX-resistant HuCC-T1 cells were also improved as shown in Figure 7, 8 and 9. In our results, ChitoHISss nanoparticles have similar tendency compared to the results of other reports [55-58]. In our results, intracellular uptake of free DOX was decreased in DOX-resistant HuCC-T1 cells while ChitoHISss nanoparticles maintained superior intracellular delivery capacity both normal HuCC-T1 cells and DOX-resistant HuCC-T1 cells (Figure 7). DOX-incorporated ChitoHISss nanoparticles showed superior anticancer activity against DOX-resistant HuCC-T1 cells while free DOX revealed decreased anticancer activity (Figure 6). Also, DOX-incorporated ChitoHISss nanoparticles efficiently inhibited tumor growth compared to free DOX through tumor-specific delivery capacity (Figure 10). Our results showed that DOX-incorporated ChitoHISss nanoparticles have potential to overcome MDR through acid pH- and redox-sensitive delivery of DOX.

  1. Gref, R.; Minamitake, Y.; Peracchia, M.T.; Trubetskoy, V.; Torchilin, V.; Langer, R. Biodegradable long-circulating polymeric nanospheres. Science. 1994, 263, 1600-1603.
  2. Lee, H.M.; Jeong, Y.I.; Kim, D.H.; Kwak, T.W.; Chung, C.W.; Kim, C.H.; Kang, D.H. Ursodeoxycholic acid-conjugated chitosan for photodynamic treatment of HuCC-T1 human cholangiocarcinoma cells. J. Pharm. 2013, 454, 74-81.
  3. Lee, H.L.; Hwang, S.C.; Nah, J.W.; Kim, J.; Cha, B.; Kang, D.H.; Jeong, Y.I. Redox- and pH-responsive nanoparticles release piperlongumine in a stimuli-sensitive manner to inhibit pulmonary metastasis of colorectal carcinoma cells. Pharm. Sci. 2018, 107, 2702-2712. 
  4. Lee, S.J.; Jeong, Y.I. Hybrid nanoparticles based on chlorin e6-conjugated hyaluronic acid/poly(l-histidine) copolymer for theranostic application to tumors. Mater. Chem. B. 2018, 6, 2851-2859.
  5. Curry, J.M.; Sprandio, J.; Cognetti, D.; Tumor microenvironment in head and neck squamous cell carcinoma. Oncol. 2014, 41, 217-234.
  6. Catalano, V.; Turdo, A.; Di Franco, S.; Dieli, F.; Todaro, M.; Stassi, G. Tumor and its microenvironment: a synergistic interplay. Cancer Biol. 2013, 23, 522-532.
  7. Wu, Y.; Zhang, X.; Feng, H.; Hu, B.; Deng, Z.; Wang, C.; Liu, B.; Luan, Y.; Ruan, Y.; Liu, X.; et al. Exploration of redox-related molecular patterns and the redox score for prostate cancer. Med. Cell. Longev. 2021, 2021, 4548594.
  8. Jorgenson, T.C.; Zhong, W.; Oberley, T.D. Redox imbalance and biochemical changes in cancer. Cancer Res. 2013, 73, 6118–6123.
  9. Correia, A.L.; Bissell, J. The tumor microenvironment is a dominant force in multidrug resistance. Drug Resist. Updat. 2012, 15, 39-49.
  10. Wu, P.; Gao, W.; Su, M.; Nice, E.C.; Zhang, W.; Lin, J.; Xie, N. Adaptive mechanisms of tumor therapy resistance driven by tumor microenvironment. Cell Dev. Biol. 2021, 9, 641469.
  11. Erin, N.; Grahovac, J.; Brozovic, A.; Efferth, T. Tumor microenvironment and epithelial mesenchymal transition as targets to overcome tumor multidrug resistance. Drug Resist. Updat. 2020, 53, 100715. 
  12. Wang, J.X.; Choi, S.Y.C.; Niu, X.; Kang, N.; Xue, H.; Killam, J.; Wang, Y. Lactic acid and an acidic tumor microenvironment suppress anticancer immunity. J. Mol. Sci. 2020, 21, 8363.
  13. Liu, Y.; Li, Q.; Zhou, L.; Xie, N.; Nice, E.C.; Zhang, H.; Huang, C.; Lei, Y. Cancer drug resistance: redox resetting renders a way. Oncotarget. 2016, 7, 42740-42761.
  14. Welters, M.J.; Fichtinger-Schepman, A.M.; Baan, R.A.; Flens, M.J.; Scheper, R.J.; Braakhuis, B.J. Role of glutathione, glutathione S-transferases and multidrug resistance-related proteins in cisplatin sensitivity of head and neck cancer cell lines. J. Cancer. 1998, 77, 556-561.
  15. Nunes, S.C.; Serpa, J. Glutathione in ovarian cancer: A double-edged sword. J. Mol. Sci. 2018, 19, 1882.
  16. Hwang, J.H.; Choi, C.W.; Kim, H.W.; Kim, D.H.; Kwak, T.W.; Lee, H.M.; Kim, C.H.; Chung, C.W.; Jeong, Y.I.; Kang, D.H. Dextran-b-poly(L-histidine) copolymer nanoparticles for pH-responsive drug delivery to tumor cells. J. Nanomedicine. 2013, 8, 3197-3207
  17. García, M.C.; Calderón-Montaño, J.M.; Rueda, M.; Longhi, M.; Rabasco, A.M.; López-Lázaro, M.; Prieto-Dapena, F.; González-Rodríguez, M.L. pH-temperature dual-sensitive nucleolipid-containing stealth liposomes anchored with PEGylated AuNPs for triggering delivery of doxorubicin. J. Pharm. 2022, 619, 121691.
  18. Palanikumar, L.; Al-Hosani, S.; Kalmouni, M.; Nguyen, V.P.; Ali, L.; Pasricha, R.; Barrera, F.N.; Magzoub, M. pH-responsive high stability polymeric nanoparticles for targeted delivery of anticancer therapeutics. Commun Biol. 2020, 3. 95.
  19. Ling, X.; Tu, J.; Wang, J.; Shajii, A.; Kong, N.; Feng, C.; Zhang, Y.; Yu, M., Xie, T.; Bharwani, Z.; Aljaeid, B.M.; Shi, B.; Tao, W.; Farokhzad, O.C. Glutathione-responsive prodrug nanoparticles for effective drug delivery and cancer therapy. ACS Nano. 2019, 13, 357-370.
  20. Li, W.; Chen Z.; Liu, X.; Lian, M.; Peng, H.; Zhang, C. Design and evaluation of glutathione responsive glycosylated camptothecin nanosupramolecular prodrug. Drug Deliv. 2021, 28, 1903-1914.
  21. Chen, R.; Ma, Z.; Xiang, Z.; Xia, Y.; Shi, Q.; Wong, S.C.; Yin, J. Hydrogen peroxide and glutathione dual redox-responsive nanoparticles for controlled DOX release. Biosci. 2020, 20, e1900331. 
  22. Yoon, H.M.; Kang, M.S.; Choi, G.E.; Kim, Y.J.; Bae, C.H.; Yu, Y.B.; Jeong, Y.I. Stimuli-responsive drug delivery of doxorubicin using magnetic nanoparticle conjugated poly(ethylene glycol)-g-chitosan copolymer. J. Mol. Sci. 2021, 22, 13169. 

  1. All acronyms have to be explained at first use in the text (The abstract stands separated).

Answer) Thanks for your comment. According to your comment, all acronyms were explained in the text (independently to Abstract section).

  1. The authors have to provide the number of measurements performed for each test (n) in sections 2.10, 2.11, 2.12, Table 1, Figs. 4,5,6,8,10.

Answer) Thanks for your comment. According to your comment, we added the number of measurements in the Figure captions and Experimental section.

In Experimental section

2.11. Anticancer activity of DOX-incorporated ChitoHISss nanoparticles against DOX-resistant HuCC-T1 cells

To assess anticancer activity of DOX-incorporated ChitoHISss nanoparticles, HuCC-T1 cells (1×104 cells/well) seeded in 96 well plates were incubated overnight in 5% CO2 at 37oC. For DOX treatment, free DOX was dissolved in DMSO and diluted with media. For nanoparticle treatment, aqueous solution of DOX-incorporated ChitoHISss nanoparticles were sterilized with 1.2 µm syringe filter and then diluted with media. 0.5 % (v/v) DMSO was used for control treatment. Cells were exposed to free DOX, DOX-incorporated ChitoHISss nanoparticles and empty nanoparticles for 1 day or 2 days. Following this, viability of cells was evaluated with MTT proliferation assay. MTT solution (30 µl, 5 mg/ml in PBS) was added to cells in 96 wells and then incubated for 3 h. Supernatants were discarded and then DMSO (100 µl) was added to dissolve viable cells and then measured absorbance at 570 nm using Infinite M200 pro microplate reader. Each measurement was average ± standard deviation (S.D.) from eight wells of 96 well plates.   

2.12. Observation of cells with fluorescence microscope

For fluorescence observation of cells, 3 × 105 HuCC-T1 cells were seeded in 6 wells with cover glass. These were treated with free DOX or DOX-incorporated ChitoHISss nanoparticles for 60 min. After that, cells were washed with PBS, fixed with 4 % paraformaldehyde for 15 min, washed again with PBS again and then immobilized with mounting solution (Immunomount, Thermo Electron Co., Pittsburgh, PA, USA). Cells were observed with fluorescence microscope (Eclipes 80i; Nikon, Tokyo, Japan). Each measurement from fluorescence observations and analysis was at least repeated three times and then presented as an average image.

2.13. In vivo animal study

Tumor xenograft model of HuCC-T1 cells was prepared to study antitumor activity of DOX-incorporated ChitoHISss nanoparticles. 1 × 107 HuCC-T1 cells were subcutaneously injected into the backs of male nude mice. Male nude mice (4-5 weeks old, 20 – 25 g) (Orient, Seongnam, Gyeonggido, Korea) were used for animal study. Five male mice were used for each group. Free DOX solution, DOX-incorporated ChitoHISss nanoparticles and empty nanoparticles were intravenously (i.v.) injected via tail vein of mice when diameter of tumor mass was reached approximately 4–5 mm. Injection volume was 100 µl. Treatment dose as a DOX was adjusted to 10 mg/kg. 5 mice were used for each group. Tumor volume and body weight were measured at intervals of 5 days. The day of drug injection was determined as a first day. Largest and smallest diameter of the tumor were measured and then tumor volume was evaluated using following formula: V = (a × [b]2)/2. a, largest diameter; b, smallest diameter. All results were expressed as average ± S.D. from five mice.

For fluorescence imaging of animals, 1 × 107 HuCC-T1 cells were subcutaneously injected into the backs of male nude mice. When diameter of tumor mass became larger than 6 mm, Ce6-incorporated nanoparticles were intravenously (i.v.) injected to the tail vein (10 mg/kg as a Ce6 concentration) of mice. Injection volume was 100 µl. 1 day later, mice were anesthetized with avertin to observe fluorescence imaging of HuCC-T1 tumor. After that, mice were sacrificed for observation of each organ. MaestroTM 2 small animal imaging instrument (Cambridge Research and Instruments, Inc. Woburn, MA, USA) was used for observation of biodistribution of nanoparticles. Each measurement from fluorescence observations and analysis was at least repeated three times and then presented as an average image.

In Figure captions

Figure 4. Fluorescence emission spectra of Ce6-loaded ChitoHISss nanoparticles. (a) The effect of pH; (b) The effect of GSH. Ce6-incorporated ChitoHISss nanoparticles were incubated with various pHs and/or GSH for 4 h. Fluorescence analysis was at least triplicated.

Figure 5. DOX release from ChitoHISss nanoparticles. (a) The effect of drug contents. (b) The effect of pH of the nanoparticle solution. (c) The effect of GSH addition and acidic pH. All results were triplicated and expressed as average ± S.D.

Figure 6. Cell cytotoxicity of DOX or DOX-incorporated ChitoHISss nanoparticles against HuCC-T1 cells and DOX-resistant HuCC-T1 cells. (a) DOX; (b) DOX-incorporated ChitoHISss nanoparticles; (c) Empty nanoparticles. Each measurement was average ± standard deviation (S.D.) from eight wells of 96 well plates.

Table 2. IC50 value of DOX or DOX-incorporated ChitoHISss nanoparticles

IC50 (µg/ml) a

HuCC-T1 cells

DOX-resistant HuCC-T1 cells

DOX

DOX-ChitoHISss NP

Emp NP

0.28 ± 0.013

0.32 ± 0.012

> 100

> 10

0.68 ± 0.024

> 100

a IC50 value was estimated from results of Figure 5. DOX-ChitoHISss NP: DOX-incorporated ChitoHISss nanoparticles; Empty NP: empty nanoparticles.

All results were triplicated and expressed as average ± S.D.

Figure 8. Cell cytotoxicity of DOX and DOX-incorporated ChitoHISss nanoparticles against DOX-resistant HuCC-T1 cells. The effect of media pH (a) and addition of GSH (b). For cytotoxicity study, cells were exposed to each pH solutions for 6 h and then media was replaced with normal serum-free media. After that, cells were further cultured for 24 h. DOX-ChitoHISss NP: DOX-incorporated ChitoHISss nanoparticles. DOX-concentration: 0.1 µg/ml. Each measurement was average ± standard deviation (S.D.) from eight wells of 96 well plates.

Figure 10. Antitumor activity of DOX-incorporated ChitoHISss nanoparticles. (a) Tumor growth; (b) body weight changes. Tumor xenograft model of DOX-resistant HuCC-T1 cells was prepared in the back of nude BALb/C mice. DOX-incorporated nanoparticles were i.v. administered through tail vein (Injection volume: 100 µl; dose, 10 mg/kg as a DOX concentration). Treatment dose as a DOX was adjusted to 10 mg/kg. All results were expressed as average ± S.D. from five mice. (c) Fluorescence imaging of tumor-xenograft model of DOX-resistant HuCC-T1 cells. For fluorescence image, Ce6-incorporated ChitoHISss nanoparticles were i.v. administered via tail vein of mouse (10 mg/kg as a Ce6 concentration). Injection volume was 100 µl. 1 day later, mice were sacrificed for observation of each organ. Empty NP: empty nanoparticle; DOX-ChitoHISss NP: DOX-incorporated nanoparticles. Control vs. DOX, p < 0.01; Control vs. DOX-ChitoHISss NP, p < 0.005; Empty NP vs. DOX, p < 0.01; Empty NP vs. DOX-ChitoHISss NP, p < 0.005.

  1. For statistical analysis, the authors have to use ANOVA instead of Student’s test, allowing comparison of data from all analyzed groups. Statistical significance has to be presented for all obtained results that were compared.

Answer) Thanks for your comment. According to your comment, We did ANOVA test instead of Students t-test and then revised the manuscript.

Experimental part

2.15. Statistical analysis

A one-way analysis of variance (ANOVA) followed by the Tukey test was employed to analyse the statistical significance using GraphPad Prism 9 (GraphPad Software LLC., San Diego, CA, USA). p < 0.05 as the minimum of significance was evaluated.

Results part

Figure 5. DOX release from ChitoHISss nanoparticles. (a) The effect of drug contents. (b) The effect of pH of the nanoparticle solution. (c) The effect of GSH addition and acidic pH. All results were triplicated and expressed as average ± S.D. Statistical analysis: * indicates comparison between 8.5 % and 13.4 %; * indicates also comparison between pH 7.4 and pH 6.8 or pH 6.0.; ** indicates comparison between GSH (0 mM) and GSH 1.0 mM, 5 mM, 10 mM or GSH 10 mM/pH 6.0. ANOVA followed by Turky’s test, p < 0.05.

Figure 6. Cell cytotoxicity of DOX or DOX-incorporated ChitoHISss nanoparticles against HuCC-T1 cells and DOX-resistant HuCC-T1 cells. (a) DOX; (b) DOX-incorporated ChitoHISss nanoparticles; (c) Empty nanoparticles. Each measurement was average ± standard deviation (S.D.) from eight wells of 96 well plates. * indicates comparison between HuCC-T1 and DOX-resistant HuCC-T1. ANOVA followed by Turky’s test, p < 0.05.

Figure 8. Cell cytotoxicity of DOX and DOX-incorporated ChitoHISss nanoparticles against DOX-resistant HuCC-T1 cells. The effect of media pH (a) and addition of GSH (b). For cytotoxicity study, cells were exposed to each pH solutions for 6 h and then media was replaced with normal serum-free media. After that, cells were further cultured for 24 h. DOX-ChitoHISss NP: DOX-incorporated ChitoHISss nanoparticles. DOX-concentration: 0.1 µg/ml. Each measurement was average ± standard deviation (S.D.) from eight wells of 96 well plates. Each measurement was average ± standard deviation (S.D.) from eight wells of 96 well plates. * indicates comparison between DOX and DOX-ChitoHISss NP. ANOVA followed by Turky’s test, p < 0.05.

Figure 10. Antitumor activity of DOX-incorporated ChitoHISss nanoparticles. (a) Tumor growth; (b) body weight changes. Tumor xenograft model of DOX-resistant HuCC-T1 cells was prepared in the back of nude BALb/C mice. DOX-incorporated nanoparticles were i.v. administered through tail vein (Injection volume: 100 µl; dose, 10 mg/kg as a DOX concentration). Treatment dose as a DOX was adjusted to 10 mg/kg. All results were expressed as average ± S.D. from five mice. (c) Fluorescence imaging of tumor-xenograft model of DOX-resistant HuCC-T1 cells. For fluorescence image, Ce6-incorporated ChitoHISss nanoparticles were i.v. administered via tail vein of mouse (10 mg/kg as a Ce6 concentration). Injection volume was 100 µl. 1 day later, mice were sacrificed for observation of each organ. Empty NP: empty nanoparticle; DOX-ChitoHISss NP: DOX-incorporated nanoparticles. * indicates comparison between Control and DOX solution; ** indicates comparison between control and DOX-ChitoHISss NP. ANOVA followed by Turky’s test, p < 0.05.

  1. The authors should provide the full origin of the reagents, apparatuses and software used (company, city, [state], country).

Answer) Thanks for your comment. According to your comment, we indicated the origin of the reagents, apparatus and software. Thanks again.

In Experimental section

Chitosan oligosaccharide (COS) was purchased from Tokyo Chemical Industry (TCI) Co., LTD (Tokyo, Japan). Doxorubicin (DOX) was purchased from LC Labs® Co. (Woburn, MA 01801, USA). Chlorin e6 (Ce6) was obtained from Frontier Sci. Co. (Logan, UT. USA). Pyrene, L-histidine methyl ester dihydrochloride (HIS), 3,3’-dithiodipropionic acid di(N-hydroxysuccinimide ester) (DTP-NHS), L-glutathione reduced (GSH), tribromoethanol (avertin), triethylamine (TEA), 3-(4,5-dimethyl-2-thiazolyl)-2, 5-diphenyl-2H-tetrazolium bromide (MTT), dimethyl sulfoxide (DMSO) and methanol (MeOH) were purchased from Sigma Aldrich Chem. Co. (St. Louis, Missouri, USA). Dialysis membranes (Molecular weight cutoffs (MWCO): 1000 and 2000 Da) were purchased from Spectrum Labs., Inc. (Rancho Dominguez, CA, USA). Organic solvents such as DMSO and MeOH was used as a ultrapure grade.

Zetasizer Nano-ZS® (Malvern, Worcestershire, UK)

fluorescence spectrofluorophotometer (Shimadzu RF-5301PC spectrofluorophometer, Kyoto, Japan)

Maestro 2 small animal imaging instrument (Cambridge Research and Instrumentation Inc., MA 01801, USA).

Falcon® tube (Thermo Fisher Sci., Co., Waltham, MA, USA)

shaker incubator (SI-600R, Jeiotech Co., Daejeon, Korea)

UV spectrophotometer (UV-1601 UV-VIS spectrophotometer, Shimadzu, Kyoto, Japan).

RPMI1640 medium (Gibco, Grand Island, NY, USA)

FBS (Invitrogen, Waltham, MA, USA

96 well plates (SPL Life Sci., Pocheon-si, Gyeonggi-do, South Korea)

1.2 µm syringe filter (Minisart® Syringe filter, Sartorius AG, Göttingen, Land Niedersachsen, Germany)

mounting solution (Immunomount, Thermo Electron Co., Pittsburgh, PA, USA).

fluorescence microscope (Eclipes 80i; Nikon, Tokyo, Japan)

GraphPad Prism 9 (GraphPad Software LLC., San Diego, CA, USA)

  1. In section 3.3, the authors have to mention that Dox-ChitoHISss reduced dramatically cell viability.

Answer) Thanks for your comment. According to your comment, we discussed again for this result in Results section and Discussion section.

In results section.

Especially, cell viability was dramatically decreased according to the concentration of DOX-incorporated ChitoHISss nanoparticles (Figure 6(b) compared to those of DOX itself against HuCC-T1 cells or DOX-resistant HuCC-T1 cells (Figure 6(a)). These results might be due to that DOX-incorporated ChitoHISss nanoparticles can be easily entered into intracellular compartment both HuCC-T1 cells or DOX-resistant HuCC-T1 cells and then killed the cancer cells.

Figure 7 supported the results of Figure 6, i.e. red fluorescence intensity of DOX-resistant HuCC-T1 cells was significantly decreased at DOX treatment compared to HuCC-T1 cells. These results indicated that DOX uptake by cancer cells was inhibited when DOX itself was treated to cells because HuCC-T1 cells became resistant to DOX and then DOX has difficulties in intracellular delivery against DOX-resistant HuCC-T1 cells as shown in Figure 7(a) and (b). These behaviors affected to the cell viability curves as shown in Figure 6(a) and (b). When DOX-incorporated ChitoHISss nanoparticles was treated, fluorescence intensity was not significantly decreased at DOX-resistant HuCC-T1 cells compared to HuCC-T1 cells as shown in Figure 7(a) and (b), indicating that DOX-incorporated ChitoHISss nanoparticles can be delivered to intracellular compartment of DOX-resistant HuCC-t1 cells and then killed cancer cells as well as HuCC-t1 cells.

In Discussion section

However, DOX-incorporated ChitoHISss nanoparticles showed almost similar tendency in cell viability, i.e. viability of DOX-resistant HuCC-T1 cells were dose-dependently inhibited by treatment of DOX-incorporated ChitoHISss nanoparticles as shown in Figure 6(b). Figure 7 supported these results, i.e. significant decrease of fluorescence intensity in cancer cells was observed at treatment of DOX itself while those were not significantly changed by treatment of DOX-incorporated ChitoHISss nanoparticles. These results indicated that DOX-incorporated ChitoHISss nanoparticles are easily entered into the intracellular compartment of cancer cells and then kill cancer cells while uptake of DOX itself was relatively inhibited by cancer cells. These results indicated that ChitoHISss nanoparticles have potential to overcome MDR of CCA tumor.

Other general minor points to be solved:

The most important general point to be solved is related to the English language. An extensive correction of grammar errors is required, especially for Introduction, but many other errors (including typing errors) appear in all the manuscript. Also, many sentences have to be rephrased. In the Materials and Methods and in the Results sections the authors have to use Past Tense. Few examples of problems to be corrected:

Answer) Thanks for your comment. According to your comment, we corrected the expression of English and revised the manuscript.

- please rephrase sentences (including the use Past Tense) in lines: 24,27,28,41,49,50,52-54,55-56,64,65-66,67,69,70-71,73-74,77,84,92,94,96-97,98-99,100-102,125,140-141,145,151-152,152-153,164,180,187-188,191,197,198,202-203,208`,217,218,220,231,257,281,288,330,352-353,357,367,369,377,380,380-381,382-383,384,386,386,390-391,393,395,414,415,418,419,422,425,443,447,448,449,450-451,459,484-485,487,492,537,541,551;

Answer) Thanks for your comment. According to your comment, we corrected these parts.

- in lines 57 and 61 please remove coma between subject and predicate;

Answer) Thanks for your comment. According to your comment, we corrected these parts.

- [21] should be replaced with [19] in line 75, and [19] should be replaced with [21] in line 76

Answer) Thanks for your comment. According to your comment, we corrected and revised these parts.

For example, Massa et al. reported that paclitaxel-incorporated albumin nanoparticles have potential to overcome MDR and then delay tumor growth/vasculature [19]. Chakrabarti et al. also reported that drug resistant problem of fibroblast growth factor receptor (FGFR) inhibitors has also to be solved for future trials [21].

- in line 87: “known to be acidic less than pH 7.0” is a pleonasm. There is no acidic pH over pH 7.0. Also, please replace “peculiarity” with another, more appropriate word in the same line;

Answer) Thanks for your comment. According to your comment, we corrected and revised these parts.

Especially, acid pH of tumor microenvironment has been applied to control drug delivery behavior of nanocarriers in tumor tissues [31].

- typing errors: lines 23 (a infrared), 90 (acidic H), 93 (rug), 97 (sun), 245 (2.11), 348 (increased);

Answer) Thanks for your comment. According to your comment, we corrected these parts.

- the word “Furthermore” is overused.

Answer) Thanks for your comment. According to your comment, we minimized the use of word “Furthermore” in the manuscript.
